

# Contrasting transit times of water from peatlands and eucalypt forests in the Australian Alps determined by tritium: implications for vulnerability and the source of water in upland catchments

5   **I. Cartwright[1,2] and U. Morgenstern[3]**

[1] *School of Earth, Atmosphere and Environment, Monash University, Clayton, Vic. 3800, Australia*

[2] *National Centre for Groundwater Research and Training, GPO box 2100, Flinders University,*
10   *Adelaide, SA 5001, Australia*

[3] *GNS Science, Lower Hutt 5040, New Zealand*

Correspondence to: I. Cartwright (ian.cartwright@monash.edu)



## Abstract

Peatlands are a distinctive and important component of many upland regions that commonly contain

distinctive flora and fauna that are different from adjacent forests and grasslands. Peatlands also

represent a significant long-term store of organic carbon. While their environmental importance has

5    long since been recognised, water transit times within peatlands are not well understood. This study

uses $^3$H to estimate the mean transit times of water from peatlands and from adjacent gullies that

contain eucalypt forests in the Victorian Alps (Australia). $^3$H activities of the peatland water range from

2.7 to 3.3 TU, which overlap the measured (2.9 to 3.0 TU) and expected (2.8 to 3.2 TU) average $^3$H

activities of rainfall in this region. Even accounting for seasonal recharge by rainfall with higher $^3$H

10   activities, the mean transit times of the peat waters are <6.5 years and most waters probably have

mean transit times of less than 2 years. Water from adjacent eucalypt forest streams has $^3$H activities

of 1.6 to 2.1 TU, implying much longer mean transit times of 5 to 29 years. Cation/Cl and Si/Cl ratios

are higher in the eucalypt forest streams than the peatland water and both have higher cation/Cl and

Si/Cl ratios than rainfall. The major ion geochemistry reflects the degree of silicate weathering in these

15   catchments that is controlled by both transit times and aquifer lithology. The short transit times imply

that, unlike the eucalypt forests, the peatlands do not represent a long-lived store of water to the local

river systems. Additionally, the peatlands are susceptible to drying out during drought which renders

them vulnerable to damage by the periodic bushfires that occur in this region.



## 1. Introduction

Globally peatlands occupy parts of the upland reaches of many river catchments and provide water to the headwater streams in those catchments. It is estimated that peatlands cover ~$4\times10^6$ km² or ~3% of the world's land area (Dixon et al., 1994). Peatlands have received much attention because they

represent a major (1.8 to $4.5\times10^{11}$ tonne) long-lived store of terrestrial organic carbon (Gorham, 1991; Page et al., 2002). Peatlands commonly contain distinctive flora such as sphagnum moss, sedges, and orchids that may not occur in surrounding forests or grasslands (Grover and Baldock, 2013; Grover et al., 2005; McDougall, 1982); due to their unique flora, there may be also differences in fauna between the peatlands and the surrounding regions.

Draining of peatlands is a major environmental concern. This can be directly due to anthropogenic activities such as peat extraction and the conversion of peatland to agricultural land or forest. Additionally, since peatlands form in high rainfall environments, drainage can occur due to a reduction in rainfall. Draining of peatlands destroys the habitats of flora and fauna and also causes oxidation of the organic matter, which in turn releases $CO_2$ to the atmosphere (Gorham, 1991; Grover and Baldock,

2010). In addition, the dried peat is susceptible to bushfires that commonly occur during droughts and which can destroy the peatlands (Van Der Werf et al., 2010).

Peatlands comprise ~520 km² or ~2.5% of the alpine region of the southeast Australian mainland (Fig. 1) (Costin, 1972; Grover et al., 2005; Western et al., 2009). The peat generally occupies shallowly-sloping areas in the upland plains that are poorly drained. While it forms only a minor part of the

Australian landscape, understanding the water balance in the Australian peatlands is important for assessing the potential impacts of environmental change. If water residence or transit times are short, then the peatlands may dry significantly during droughts making it prone to degradation and bushfires. Additionally, the peatlands provide water to numerous small upland streams. Most of the upland peatlands are flanked by eucalypt forests that occupy steeper slopes and gullies. Eucalypts have high

transpiration rates and groundwater recharge rates in eucalypt-dominated areas are low (Allison et al., 1990), which results in relatively long water transit times (on the order of years to decades) in eucalypt-dominated catchments (Cartwright and Morgenstern, 2015).



## 1.1 Determining mean transit times

The mean transit time represents the average time required for water to flow through the aquifer or soil from where it recharges to where it discharges at the surface (e.g., at a spring, stream or lake) or is sampled from a bore (Cook and Bohlke, 2000; Maloszewski and Zuber, 1982; McDonnell et al., 2010).

Documenting mean transit times is important for understanding and managing catchments. The time required for nutrients or contaminants to be transported from recharge areas to streams is a direct function of the transit time (Morgenstern and Daughney, 2012). Additionally, catchments with long transit times are more likely to be resilient to short-term (years to decades) variations in rainfall but will respond to climate or landuse changes that cause longer-term (decades to centuries) changes in

groundwater recharge and flow.

In the southern hemisphere, $^3$H is an ideal tracer for determining transit times of shallow groundwater, soil water, or surface water (Morgenstern et al., 2010). Since it is part of the water molecule, $^3$H activities are controlled only by the initial activities in the recharging water and the time in the flow system. Other potential tracers for determining transit times of young waters, such as $^3$He,

chlorofluorocarbons, and $SF_6$ (Cook and Bohlke, 2000) are gases that degas to the atmosphere and are thus difficult to use for water sampled from streams. Stable isotope ratios or Cl concentrations may be also used to estimate mean transit times (Hrachowitz et al., 2010; Kirchner et al., 2010). However, this is only practicable where mean transit times are shorter than ~5 years, as longer transit times attenuates the seasonal variation of these tracers in the rainfall (Stewart et al., 2010). Additionally,

use of these tracers requires a detailed (preferably at least weekly) record of stable isotope and/or major ion geochemistry in rainfall for a period which exceeds that of the transit times of water in the catchment.

Coupled with models that describe the distribution of flow paths through an aquifer (Cook and Bohlke, 2000; Maloszewski, 2000; McGuire and McDonnell, 2006), $^3$H may be used to determining transit

times of waters that are up to ~100 years old. The $^3$H activities in rainfall over the last several decades have been measured globally (e.g. International Atomic Energy Association, 2016; Tadros et al., 2014), which allows the $^3$H input to catchments over time to be estimated. Rainfall $^3$H activities peaked in the 1950s to 1960s due to the production of $^3$H in atmospheric thermonuclear tests (commonly termed




"bomb pulse" $^3$H). The $^3$H activities of remnant bomb pulse waters in the northern hemisphere are currently above those of modern rainfall, which results in individual measurements of $^3$H activities in groundwater or surface water yielding non-unique estimates of transit times (Morgenstern et al., 2010). However, the bomb pulse $^3$H activities in the southern hemisphere were several orders of

magnitude lower (Clark and Fritz, 1997; Tadros et al., 2014), and have now decayed below those of modern rainfall. This permits unique transit times to be estimated from single $^3$H measurements (Morgenstern et al., 2010). Because the $^3$H activities of the remnant bomb pulse waters are below those of modern rainfall, relative transit times are independent of assumed flow models (i.e., water with low $^3$H activities has longer mean transit times than water with high $^3$H activities).

**1.2 Transit times in upland peat**

The transit time of peatland water, especially upland or mountain peat, is poorly known. Due to peatlands containing poorly-drained organic rich soils and occupying shallow slopes, it is sometimes assumed that peatland water has long transit times. However, there has been little assessment of that assumption (e.g., as discussed by Western et al., 2009). Using $^3$H activities, transit times of years to

decades were proposed for water from metre to tens-of-metre thick peat deposits in Quebec, Canada (Dever et al., 1984), Dartmoor, UK, (Charman et al., 1999), Lithuania (Mažeika et al., 2009), and Minnesota, USA (Siegel et al., 2001). From the preservation of seasonal $\delta^{18}$O and $\delta^2$H values, Aravena and Warner (1992) estimated that the water from the upper ~0.1 m of peat from Ontario, Canada, was <1 year old.  Mean transit times from the extensively-studied Plynlimon, Feshie, Mharcaidh and

Girnock catchments, UK, which include regions of peat, were estimated using the stable isotope and Cl variation in stream water as <5 years (Benettin et al., 2015; Kirchner et al., 2010; Soulsby et al., 2006). However, the mean transit times of the water specifically draining the peat in those catchments is not known. Morris and Waddington (2011) modelled water transit times in peat and suggested that they vary from <1 year for the upper 0.1 m to several years in deeper layers, which is broadly

consistent with the above estimates. While not estimating transit times, Western et al. (2009) concluded on the basis of hydrograph analysis and water balance calculations that peatlands in southeast Australia did not act as long-term water stores for the local streams.





### 1.3 Objectives

This study is based in the upland areas of the Victorian Alps, southeast Australia and was designed to test the hypothesis that peatlands represented a relatively long-lived store of water in the upper catchments. Specifically, we utilise $^3$H to provide the first estimates of the mean transit times of water draining Australian peatlands and compare these with mean transit times of streams that drain adjacent eucalypt forests. Documenting mean transit times is important for determining the relative importance of the peatlands and eucalypt forests in providing water to the local rivers and also in assessing potential environmental impacts resulting from landuse or climate change. In addition, we assess whether the major ion geochemistry of the water may be used as a first-order proxy for mean transit times and discuss the controls on mean transit times in these upland areas.

Mean transit times in adjacent catchments in the Ovens Valley dominated by eucalypt forests (Fig. 1) were estimated using $^3$H as being between 4 and 30 years, with higher mean transit times recorded during summer low flow conditions (Cartwright and Morgenstern, 2015). That study focussed on the major tributaries to the Ovens River that had catchment areas between 6 and 435 km$^2$, and the transit times in the upper reaches of the catchments where streamflow commences remains unknown. Additionally, it was not established whether there are differences between mean transit times for water draining the peatlands and the eucalypt forests in those upper catchments.

### 2. Setting

Peatland water and eucalypt forest streams were sampled in the Mount Buffalo (Ovens Catchment), Falls Creek (Kiewa and Upper Murray Catchments), and Dargo (Mitchell Catchment) regions of the Victorian Alps (Fig. 1). Catchment areas in the peatlands and eucalypt forests are similar, ranging from 0.5 to 6.4 km$^2$ (Table 1). The Falls Creek and Dargo areas consist of indurated Ordovician metasedimentary rocks, Devonian granites, and minor Tertiary basalts, whereas the Mount Buffalo plateau comprises Devonian granite (van den Berg et al., 2004; Energy and Earth Resources, 2016).

Peatlands are developed in poorly-drained low-relief upland areas with eucalyptus forests occupying the steeper valleys dissecting the uplands and the higher better-drained areas at the margins of the peatland (Grover and Baldock, 2013; Grover et al., 2005; McDougall, 1982; Western et al., 2009). The



peatlands are characterised by sphagnum mosses (Sphagnum cristatum), rushes (Empodisma minus and Baloskion australe), and heaths (Epacris paludosa and Richea continentis) (McDougall, 1982). Most of the peatlands in the Alpine region are <2 m thick. The upper few centimetres comprise little decomposed plant material (the fibric zone). The fibric zone grades downwards through the hemic

zone (typically 50 cm to 1 m thick) where the soils are organic-rich, fiberous, with abundant discernible roots into a denser soil (the sapric zone) comprising mainly decomposed organic matter with clays and fragments of rock. The peat typically has a thin (<20 cm thick) layer of weathered rocks at its base but mainly overlies little-weathered basement rocks. The hydrology of the peat comprises the upper acrotelm, which alternates between saturated and unsaturated as the water table rises and falls, and

the lower permanently-saturated catotelm (e.g., Grover et al., 2005). The boundary between the acrotelm and catotelm is generally in the hemic zone.

In the alpine regions of southeast Australia the dominant eucalypt species include Mountain Ash (Eucalyptus regnans) at altitudes of <1000 m and Alpine Ash (Eucalyptus delegatensis) and Snow Gum (Eucalyptus pauciflora) at higher altitudes (McDougall, 1982). The soils of the eucalypt forests include

sandy lithosols; these are thin, well-drained, and poor in organic matter and largely occur on the upper slopes of the gullies and on the elevated areas surrounding the peatlands. In the centres of the gullies, the soils alci include alpine humus loams that contain higher contents of organic matter and which are less well drained. The soils overlie weathered regolith that is a few tens-of-centimetres to a few metres thick and there are also minor deposits of colluvium and alluvium along the streams (Shugg, 1987).

Groundwater flow is restricted to the weathered zones and fractures in the granites and metasediments; the minor alluvial and colluvial sediments are more permeable but represent only a minor part of the landscape.

Average precipitation is 1250 mm yr$^{-1}$ at Falls Creek, 1650 mm yr$^{-1}$ at Mount Hotham (near Dargo), and 1790 mm yr$^{-1}$ at Mount Buffalo (Bureau of Meteorology, 2016) (Fig. 1). Approximately 60 to 70%

of precipitation occurs in the austral autumn to winter (May to September) with a proportion of the winter precipitation occurring as snow. February and March have the lowest precipitation (4 to 6% of the annual total in each month). No currently active river gauges exist in these upland areas and previous flow measurements were only from a few streams at Falls Creek (Department of Environment,





Land, Water, and Planning, 2016). These limited records show that river flow varies seasonally with the majority of streamflow occurring in the winter months; however, flows persist over the summer periods (Fig. 2a) and the streams do not record zero flows (Fig. 2b). Water from some of the peatlands at Falls Creek drains into the Rocky Valley Reservoir (~$2.8\times10^7$ m$^3$), which is used to provide water to

the Falls Creek resort and the Kiewa Valley hydroelectric system. Water from some of the peatlands at Mount Buffalo feeds Lake Catani, which is an artificial recreational lake with a volume of ~$2\times10^6$ m$^3$.

## 3. Methods

### 3.1 Sampling and analytical techniques

Aggregated rainfall samples of several weeks to months duration were collected from a rainfall

collector at Mount Buffalo (Fig. 1). Samples of stream water from the peatlands and eucalypt forests were taken from flowing reaches. Water was also sampled from within the peat using a hand-operated siphon pump with a soft PVC inlet hose that was inserted into a rigid 5 cm diameter PVC piezometer with a ~20 cm screen pushed directly into the peat. The piezometer was inserted to the maximum possible extent (i.e., until resistance prevented it from being pushed deeper). Observations indicate

that this is at or close to the base of the peat within the hemic zone and catotelm.  At least three bore volumes of water was extracted prior to sampling.

$^3$H activities were measured using liquid scintillation spectrometry (Quantulus ultra-low-level counters) at Geological and Nuclear Sciences, New Zealand on water samples that had been vacuum distilled and electrolytically enriched (Morgenstern and Taylor, 2009). $^3$H activities are expressed in tritium

units (TU) where 1 TU represents a $^3$H/$^1$H ratio of $1\times10^{-18}$. $^3$H enrichment by a factor of 95, which results in a detection limit of 0.02 TU, and the use of deuterium-calibration for each sample results in a 1% reproducibility of the tritium enrichment. Precision (1$\sigma$) is ~1.8% at 2 TU (Table 1).

Major ion concentrations were measured at Monash University using a ThermoFinnigan ICP-OES (cations), ThermoFinnigan ICP-MS (Si), and a ThermoFischer ion chromatograph (anions). Samples for

cation and Si analysis were filtered through 0.45 μm cellulose nitrate filters and acidified to pH <2 using double-distilled 16M HNO$_3$. Samples for anion analysis were filtered but unacidified. The





precision of the major ion concentrations determined by replicate analyses of samples is ±2% and the accuracy as determined by analysis of certified water standards is ±5%.

$\delta^{18}O$ and $\delta^2H$ values of water were determined using a ThermoFinnigan DeltaPlus Advantage mass spectrometer at Monash University. A ThermoFinnigan Gas Bench was used for the $^{18}O$ analyses.

Waters were equilibrated with He-$CO_2$ at 32 $^oC$ for 24 to 48 hours and the gas subsequently analysed by continuous flow. A ThermoFinnigan H-Device was used for the $^2H$ analyses. H was produced from water via reaction with Cr at 850 $^oC$ and analysed by dual-inlet measurement. Internal standards that have been calibrated using IAEA SMOW, GISP and SLAP standards were employed to normalise the $\delta^{18}O$ and $\delta^2H$ values (following Coplen, 1988). $\delta^{18}O$ and $\delta^2H$ values are expressed in ‰ relative to V-

SMOW with a precision ($1\sigma$) determined from replicate analyses of samples of ±0.1‰ ($\delta^{18}O$) and ±1‰ ($\delta^2H$). The deuterium (D) excess is defined as $\delta^2H - 8\ \delta^{18}O$ (Clark and Fritz, 1999).

Catchment areas were estimated from Google Earth satellite images and 1:30,000 topographic map sheets on which the streams are clearly distinguished. Attempts to define catchment areas using a digital elevation model (DEM) did not reproduce the drainage pattern in the peatlands due to the low

topography relative to the DEM resolution.

### 3.2  Estimating mean transit times using $^3H$

Water flowing through aquifers or soils follows flow paths of varying length and thus the water discharging into streams or sampled from bores has a range of transit times (McGuire and McDonnell, 2006). Mean transit times were calculated using lumped parameter models (Maloszewski, 2000;

Maloszewski and Zuber, 1982; Zuber et al., 2005) as implemented in the TracerLPM Excel workbook (Jurgens et al., 2012). For steady-state groundwater flow, the $^3H$ activity in water at the catchment outlet at the time of sampling ($C_o(t)$) may be estimated using the convolution integral:

$$C_o(t) = \int_0^\infty C_i(t-\tau)\ g(\tau)\ e^{-\lambda\tau} d\tau$$

(1),



In Eq. (1), $C_i$ is the $^3$H activity of recharge, $\tau$ is the transit time, $t-\tau$ is the time when water entered the catchment, $\lambda$ is the $^3$H decay constant (0.0563 yr$^{-1}$), and $g(\tau)$ is the response function that describes the distribution of transit times in the flow system.

Several lumped parameter models were considered. Transit times in homogeneous unconfined aquifers of constant thickness where recharge is uniform, and where water from the entire aquifer discharges to the stream or is sampled by a bore, are described by the exponential flow model. Flow through aquifers where flow paths are the same length and no mixing occurs results in all water discharging to the stream at any given time having the same transit time and is described by the piston flow model. Transit times transit times in aquifers that have regions of both piston and exponential flow are described by the exponential-piston flow model, for which the response function is:

$$g(\tau) = 0 \qquad \text{for } \tau < \tau_m(1-f) \qquad (2a)$$

$$g(\tau) = (f\tau_m)^{-1} e^{-\tau/f\tau_m + 1/f - 1} \qquad \text{for } \tau > \tau_m\,(1-f) \qquad (2b),$$

where $\tau_m$ is the mean transit time and $f$ is the proportion of the aquifer volume that exhibits exponential flow. At $f = 1$, this model becomes the exponential flow model, while at $f = 0$, it becomes the piston flow model. TracerLPM specifies the ratio of exponential to piston flow as the EPM ratio (equivalent to $1/f - 1$).

The dispersion model is based on the solution to the one-dimensional advection-dispersion transport equation. The response function for the dispersion model is:

$$g(\tau) = \frac{1}{\tau\sqrt{4\pi D_P \tau/\tau_m}}\, e^{-\left(\frac{(1-\tau/\tau_m)^2}{4D_P \tau/\tau_m}\right)} \qquad (3),$$

where $D_P$ is the dispersion parameter defined as $D_P = D/(v\,x)$, where $D$ is the dispersion coefficient (m$^2$ day$^{-1}$), $v$ is velocity (m day$^{-1}$) and $x$ is distance (m). This model is generally considered to be a less realistic representation of flow systems, however it does commonly reproduce the observed distribution of tracers (Maloszewski, 2000).





## 4. Results

### 4.1 Streamflow

As noted above, very few streamflow measurements exists in these catchments. A near complete streamflow record for Watchbed Creek, which drains the peatlands at Falls Rocky A, exists for the

period between 1940 and 1986. The average discharge for that creak over that time period was $5.64 \times 10^6$ m$^3$ yr$^{-1}$ (Department of Environment, Water, Land and Planning, 2016). For the average annual rainfall of 1.25 m yr$^{-1}$ (Bureau of Meteorology, 2016), the runoff coefficient (i.e., the volume of rainfall that is exported by the stream) was 78%. This is much higher than the runoff coefficients in the eucalypt-dominated catchments of the upper Ovens Valley which range from 6 to 64% (Cartwright

and Morgenstern, 2015). That Watchbed Creek exports more rainfall than streams from the eucalypt forests is also apparent in the flow duration curves (Fig. 2b). Relative discharges from Watchbed Creek are around an order of magnitude higher than those from Simmons Creek in the upper Ovens catchment that has a slightly larger catchment area (6 km$^2$ vs. 3.5 km$^2$) and receives similar rainfall, but which drains eucalypt forest.

### 4.2 Stable isotope ratios

$\delta^{18}$O and $\delta^2$H values of the peatland water range from −8.3 and −43‰ to −5.0 and −26‰, respectively. The eucalyptus forest streams have smaller ranges of $\delta^{18}$O and $\delta^2$H values (−7.4 to −5.7‰ and −38 to −29‰, respectively) and rainfall has $\delta^{18}$O of −7.9 to −6.5‰ and $\delta^2$H values of −40 to −34‰. As is locally observed in groundwater and surface water throughout southeast Australia (Cartwright et al., 2012;

Leaney and Herczeg, 1999), the $\delta^{18}$O and $\delta^2$H values of all of the water, including the rainfall, lies to the left of the global and Melbourne meteoric water lines and defines an array with a with a slope of ~5 (Fig. 3). The median D-excess is 20‰, which is far higher than the mean D-excess of 9.6‰ for the Melbourne meteoric water line (Hughes and Crawford, 2012).

### 4.3 Tritium activities

$^3$H activities of three multi month aggregated rainfall samples from Mount Buffalo are 2.85 TU (12 month aggregated sample collected in February 2015), 2.88 TU (9 month aggregated sample collected in November 2015), and 2.99 (17 month aggregated rainfall collected in December 2013) (Table 1).





Three 2 week to 3 month aggregated samples collected in 2014 also from Mount Buffalo have [3]H activities of 2.52 to 2.90 (Table 1). The sample with the lowest [3]H activity represents mainly snow and low-temperature rainfall collected over a 2 week period in July 2014. The [3]H activities are within the expected range of [3]H activities in rainfall in this region of 2.8 to 3.2 TU (Tadros et al., 2014). Aside from the rainfall sample that represents the two weeks of winter precipitation in July 2014, there is an inverse correlation ($R^2$ = 0.69) between the [3]H activities and $\delta^2$H of rainfall (Fig. 4) and, as the rainfall with the higher $\delta^2$H values has the lower D-excess, a correlation between D-excess and [3]H activities. Rainfall from elsewhere in southeast Australia defines broadly similar trends in $\delta^{18}$O vs. $\delta^2$H values and $\delta^2$H vs [3]H values (Table 1, Figs 3, 4).

[3]H activities of the peatland water vary from 2.70 to 3.32 TU (median = 2.75 TU) at Mount Buffalo, 2.90 to 3.17 (median = 3.04 TU) at Falls Creek, and 2.47 to 3.36 (median = 2.77 TU) at Dargo (Table 1, Figs 4, 5) with an overall median [3]H activity of 2.88 TU. p values from two-tailed t tests are between 0.1 and 0.9 implying that there is no significant difference in the [3]H activities of the peatland water between the three sites and. Likewise, there are no significant differences in [3]H activities between water collected from within the peat and that collected from the streams draining the peat, or between samples collected in February 2015 and those collected in November 2015. The [3]H activities of the peatland waters are locally higher than those of the aggregated rainfall samples and several of those waters have lower $\delta^2$H values and higher D-excesses than the rainfall (Figs 3, 4).

[3]H activities of the eucalyptus forest streams are between 1.56 and 2.35 TU (median = 2.10 TU) (Table 1; Figs 4, 5), which are significantly lower (p < 0.05) than the [3]H activities of the peatland water. The [3]H activities of the eucalyptus streams are similar (p = 0.7) to those of river water from the streams in the upper Ovens catchment at low flow conditions in December 2013 and February 2014, which range from 1.63 to 2.28 TU (median = 2.09 TU) (Cartwright and Morgenstern, 2015; Fig. 5).

**4.4 Major ion geochemistry**

Cl concentrations of the peatland waters are between 0.5 and 1.1 mg l$^{-1}$, which are similar to those of rainfall from Mount Buffalo (0.8 to 1.1 mg l$^{-1}$) (Fig. 5a, Table 1). There is some variation between the peatland areas, with peatland water from Falls Creek having lower Cl concentrations (0.5 to 0.9 mg l$^-$





[1]); however, there is no significant difference between the Cl concentrations of the water extracted from within the peat and that draining the peatlands. Na concentrations of the peatland water range from 0.8 to 2.1 mg l$^{-1}$, which are higher than those of rainfall (0.6 to 1.0 mg l$^{-1}$) (Fig. 5b). The eucalypt forest streams have higher Cl (1.1 to 2.6 mg l$^{-1}$) and Na (3.2 to 5.7 mg l$^{-1}$) concentrations than any of the peatland water (Figs. 5a, 5b). Molar Na/Cl ratios range from 1.7 to 3.5 in the peatland water and 3.9 to 6.8 in the eucalypt forest streams (Fig. 5d). These Na/Cl ratios are higher than those of Buffalo rainfall, which has Na/Cl ratios of 1.0 to 1.1 that are similar to those generally recorded in southeast Australia (Blackburn and McLeod, 1983; Crosbie et al., 2012).

Molar Cl/Br ratios vary between 340 and 650 in the peatland water and between 550 and 780 in the eucalypt forest streams (Fig. 5c). These ratios are similar to those of Buffalo rainfall (570 to 650) and also similar to the ocean Cl/Br ratio of ~650 (Davis et al., 1998). Other cation/Cl ratios are also higher in the eucalypt forest streams compared with the peatland water, and the water from both these sources has higher cation/Cl ratios than rainfall. Si concentrations in Buffalo rainfall are <0.1 mg l$^{-1}$ (Fig. 5e) and molar Si/Cl ratios are ~0.1 (Fig. 5f). By contrast, Si concentrations and Si/Cl ratios in the peatland water and eucalypt forest streams are significantly higher (up to 9.8 mg l$^{-1}$ and 9.5, respectively: Figs 5e, 5f).

There are broad inverse correlations between $^3$H activities and Na, Cl, and Si concentrations and Na/Cl and Si/Cl ratios (Fig. 5). In general the differences in geochemistry between the peatland water and eucalypt forest streams are more marked than the geochemical variations within those groups. For example, the Si vs. $^3$H trend for the combined peatland water and eucalypt forest streams has a $R^2$ of 0.63, whereas $R^2$ values for the peatland water and the eucalyptus forest streams are 0.05 and 0.32, respectively. The correlations between major element concentrations and $^3$H activities are stronger in the eucalypt forest streams ($R^2$ values of 0.32 to 0.87) than in the peatland waters ($R^2$ values of 0.05 to 0.24). The trends in $^3$H activities vs. major ion concentrations or ratios in the eucalyptus forest streams from the Victorian Alps are similar to those of the upper Ovens catchment as a whole.


## 5. Discussion

The combination of major ion geochemistry, stable isotope data, and $^3$H activities allows geochemical processes and mean transit times of the peatland water and eucalyptus forest streams to be understood.

### 5.1 Modern rainfall $^3$H activities

Understanding the $^3$H activities of the water that recharges the catchments is critically important in estimating mean transit times. The three longer-term (9 to 17 month) aggregated rainfall samples have $^3$H activities of between 2.85 and 2.99 TU, which lie within the predicted $^3$H activity of rainfall in this area of 2.8 to 3.2 TU (Tadros et al., 2014). However, the observation that the peatland water has $^3$H activities that are locally up to 3.35 TU indicates that it is not possible to use the aggregated rainfall $^3$H activities as the $^3$H activity of recharge for all of the peatland waters.

There may be spatial variations in rainfall $^3$H activities, especially between the three areas. The observation, however, that peatland water at Mount Buffalo from <5 km from the rainfall gauge has a $^3$H activity of 3.32 TU suggests that this cannot be the only explanation. The higher $^3$H activities in the peatland waters potentially reflect preferential recharge by spring rainfall. Maximum $^3$H activities in Australian rainfall are recorded in early spring (August to September) as this is the time of maximum transport of water vapour with high $^3$H activities from the stratosphere to the troposphere (Tadros et al., 2014). Stratospheric moisture also has lower $\delta^2$H values and higher D-excesses than tropospheric moisture (Bony et al., 2008). Variation in the relative ratios of stratospheric to tropospheric moisture would also explain the inverse correlation between $^3$H activities and $\delta^2$H values (Fig. 4) and the positive correlation between $^3$H activities and D-excess values, which are observed in most rainfall samples from Mount Buffalo and elsewhere in southeast Australia.

The peatland waters that have lower $\delta^2$H values than those of the aggregated rainfall potentially were recharged by rainfall with a high proportion of stratospheric moisture, which would also have higher $^3$H activities than average rainfall. While it is difficult to precisely constrain the $^3$H of seasonal recharge, extending the rainfall $\delta^2$H vs. $^3$H trend defined by the Mount Buffalo rainfall samples in Fig. 4 to the lowest $\delta^2$H value of the peatland waters (−43‰) yields a $^3$H activity of ~3.4 TU, which is similar to the





highest $^3$H activities recorded in the peatland waters. This is ~14% higher than the $^3$H activities of the aggregated rainfall samples, which is well within the seasonal variation of $^3$H activities of Australian rainfall (Tadros et al., 2014).

That some of the peatland waters with high $^3$H activities have higher $\delta^2$H values is likely due to subsequent evaporation (Fig. 4). As discussed below, the major ion geochemistry also implies that evaporation has affected these waters. Evaporation at a humidity of 50 to 70%, which is characteristic of these Alpine areas in summer (Bureau of Meteorology, 2016), produces arrays of $\delta^{18}$O and $\delta^2$H values with slopes of 4 to 5 (Clark and Fritz, 1999). Thus the $\delta^{18}$O and $\delta^2$H values of the waters (Fig. 3) probably reflects both initial variations in stable isotope ratios and subsequent evaporation. Assuming that the mass-dependant isotope fractionation of $^3$H/$^1$H is approximately double that of $^2$H/$^1$H, a 10‰ increase in $\delta^2$H values would equate to a ~2% increase in $^3$H activities, which is approximately the analytical precision.

**5.2   Mean transit times**

Mean transit times were calculated using an exponential-piston flow model via Eqs (1 and 2). The soils and shallow aquifers are unconfined and likely to exhibit exponential flow below the water table; however, recharge through the unsaturated zone will most likely resemble piston flow. Mean transit times were calculated assuming an exponential-piston flow model initially with $f$ = 0.75 (EPM ratio of 0.33); a similar flow model successfully reproduces time-series of $^3$H activities in catchments in New Zealand (Morgenstern et al., 2010). Given that the water were sampled during the late spring or summer, which represent the driest months, it was assumed that there was a single store of water present rather than a mixture of recent rainfall and older water.

The annual average $^3$H activities of rainfall in Melbourne, which is ~200 km to the southwest, from the International Atomic Energy Agency Global Network of Isotopes in Precipitation program as summarised by Tadros et al. (2014) were used as the $^3$H input. Rainfall $^3$H activities reached ~62 TU in 1965 and then declined exponentially to present-day values (Tadros et al., 2014). The calculations initially adopted a present-day $^3$H activity of recharge of 3.4 TU (as discussed above). $^3$H activities of 3.4 TU were also used for recharge from prior to the atmospheric nuclear tests.





Mean transit times from the exponential piston flow model are up to 6.4 years with a median of 3.0 years (Table 2). For the range of [3]H activities in these peatland waters, there is negligible difference between the mean transit times from the different lumped parameter models (Fig. 6a). Assuming a ±2% uncertainty for the [3]H activities translates into an uncertainty in mean transit times of ±0.3 years

for a water with a [3]H activity of 3 TU. Uncertainties in calculated mean transit times mainly arise from the assumed [3]H activity of the recharging water (Fig. 6b), which as discussed above is difficult to constrain precisely. Utilising a [3]H activity of 3.0 TU for modern and pre-bomb pulse rainfall based on the [3]H activities of aggregate rainfall samples (Table 1), yields mean transit times of up to 3.9 years with a median of <1 year (Table 2). However, in this case a large number of peatland waters have [3]H

activities that are higher than the rainfall activities and for those waters, a [3]H activity of 3.0 for the recharging water is not realistic. Adopting a [3]H activity of 3.2 TU, which is the upper limit of the estimate for annual rainfall in this area (Tadros et al., 2004) yields mean transit times of up to 5.4 years with a median of 1.8 years. In this case two peatland waters (Dargo D and Buffalo Cresta A) have [3]H activities that are higher than the assumed recharge [3]H activities. Given the likelihood that seasonal

recharge has occurred and thus the [3]H activity of the recharging water may be locally variable, it is unrealistic to differentiate mean transit times of <1 year.

By contrast with the peatland waters, the eucalypt forest streams have much longer mean transit times. For the exponential-piston flow model with $f = 0.75$ and a modern and pre-bomb pulse rainfall activity of 3.0 TU (which assumes that the aggregated rainfall recharges the catchments), mean transit

times range from 5.3 to 28.6 years with a median of 9.5 years (Table 2). Adopting a [3]H activity for recharge of 3.2 TU, the mean transit times range from 6.9 to 28.8 years with a median of 10.8 years and if the [3]H activity of recharge was 3.4 TU, the mean transit times are 7.8 to 28.9 years with a median of 11.5 years. These mean transit times are within the range of those of the much larger (6 to 1240 km$^2$) catchments in the upper Ovens Valley that are also dominated by eucalypt forest. Mean transit

times of river water in December 2013 and February 2014 in the upper Ovens Valley calculated using the same exponential-piston flow model and a [3]H activity of recharge of 3.0 TU are 7 to 30 years (Cartwright and Morgenstern, 2015).





Especially as mean transit times increase the assumed $^3$H activity of the recharging water makes relatively little difference to the estimated mean transit time (Fig. 6b). Varying the modern rainfall $^3$H activities between 3.0 and 3.4 TU results in a range of mean transit times of 2.8 years for a water with a $^3$H activity of 2 TU, but 0.1 year for a water with a $^3$H activity of 1.5 TU. For the eucalypt forest streams

there are larger uncertainties in mean transit times resulting from the use of different flow models (Fig. 6a). The range in mean transit times for a water with a $^3$H activity of 2 TU calculated from the exponential flow model, the exponential-piston flow model with $f = 0.33$ and $f = 1$, and the dispersion model with $D_p = 0.1$ and $D_p = 1$ is ~4.5 years. For a water with a $^3$H activity of 1.5 TU the range in mean transit times from those models is ~13 years. Propagating the analytical uncertainty in $^3$H activities

(Table 2) also results in uncertainties in mean transit times of ±0.5 years for a water with a $^3$H activity of 2 TU.

Overall, while there are uncertainties in the calculated mean transit times, the conclusion that the mean transit times of water in the eucalypt forest catchments range from several years to decades and are significantly longer than those in the peatlands remains robust. Mean transit times of the

peatland waters are unlikely to be more than a few years and may be mainly less than 2 years. As noted above, the $^3$H activities of the water from the piezometers in the peatlands are statistically similar to the water that drains the peat. The water that drains the peat is likely derived mainly from the acrotelm (Western et al., 2009; Grover and Baldock, 2013), presuming that the piezometers which were inserted into the lower levels of the peat sampled water from the catotelm, this observation

implies that the catotelm is not a store of older water.

### 5.3  Major ion tracers

The major ion geochemistry allows the main geochemical processes to be understood and also can provide first-order estimates of mean transit times. Overall, the observation that the Cl/Br ratios in the peatland water and the eucalypt forest streams have molar Cl/Br ratios that cluster around those

of the rainfall implies that, in common with the majority of groundwater and surface water in southeast Australia, the vast majority of Cl is derived from the rainfall and concentrated by evapotranspiration (Cartwright et al., 2006; Herczeg et al., 2001). Other potential sources of Cl (such





as dissolution of halite in the unsaturated zone) produce water with high Cl/Br ratios and can thus be discounted.

The higher Cl concentrations in the eucalypt forest streams compared with that from the peatland water reflects higher net evapotranspiration rates in the eucalypt forest catchments. Some of the

increase in the concentration of the other major ions such as Na in the eucalypt forest streams and peatland water over those in rainfall will also be due to evapotranspiration. The adsorption of Br by organic matter (Gerritse and George, 1988) may locally increase Cl/Br ratios if the pool of organic matter is increasing. By contrast, a net degradation of organic matter releases Br producing lower Cl/Br ratios. The wider range of Cl/Br ratios in the peatland water may reflect that, locally, the volume

of organic matter is increasing whereas elsewhere it is degrading. The far more homogeneous Cl/Br ratios in the eucalypt forest streams indicate that either the organic matter content of those catchments is relatively stable or that the waters are better mixed and any local variations in Cl/Br ratios have been homogenised. This homogenisation is consistent with longer mean transit times in the eucalypt forest streams.

The increase in cation/Cl and Si/Cl ratios in the peatland and eucalypt forest waters over those in rainfall is interpreted to be due to silicate weathering (c.f., Herczeg and Edmunds, 2000). As rainfall contains <0.1 mg l$^{-1}$ Si, evapotranspiration will not increase Si concentrations appreciable and the vast majority of the Si will be derived from silicate weathering. The eucalypt forest streams have higher cation/Cl and Si/Cl ratios compared with the peatland water and thus records higher degrees of

weathering. This is likely due to two factors. Firstly, the eucalypt forest catchments are developed on weathered regolith and the flow of water through the regolith will promote mineral dissolution. Common minerals in the regolith include plagioclase and alkali feldspars (Cartwright and Morgenstern, 2012) and the dissolution of these will provide Si, Na, Ca, and K. By contrast, the peatlands are developed on less weathered rocks, which limits the potential for mineral dissolution; although there

clearly has been sufficient weathering to produce the elevated cation/Cl and Si/Cl ratios. There is commonly a thin (<20 cm thick) layer of regolith at the base of the peat and fragments of basement rock within the lower layers of the peat that are likely being weathered by the peatland water.




Secondly, the longer mean transit times in the eucalypt forest catchments allow the silicate weathering reactions to progress further than in the peatlands.

## 5.4 Controls on mean transit times

As with the Ovens catchment as a whole, there is no correlation between $^3$H activities and catchment areas either within or between the eucalypt forest and peatland catchments (Fig. 7). Elsewhere, inverse correlations between catchment slope and mean transit times have been documented (McGuire et al., 2005). Catchment slopes have not been calculated in this study (and would be difficult do so in the peatlands given the limitations of the DEM). However, the observation that the eucalypt forests occur in gullies that invariably have steeper slopes than the peatlands but contain water with longer mean transit times indicates that, in this case, catchment slopes cannot explain the difference in mean transit times between the two catchment types.

The difference in the mean transit times between the peatland water and the eucalypt forest streams is likely the combination of two factors. Firstly the high evapotranspiration rates of the eucalypt forests (Allison et al., 1990) results in low recharge rates that in turn increase the transit times. Secondly, differences in the depth of weathering in the catchments and the development of deeper flow paths is also likely to be important. Most of the peatlands are developed on relatively unweathered basement rocks and the majority of the water is stored within the peat itself (Grover and Baldock, 2013; Western et al., 2009). Given that most of the peat is <2 m thick, this results in a shallow reservoir of water underlain by bedrock with very low hydraulic conductivities. By contrast, groundwater in the eucalypt gullies is partially hosted in weathered regolith that is locally several metres thick. The gullies may be developed where weathering is greater due to the presence of joints and/or faults that will also host groundwater flow (Shugg, 1987). The presence of weathered and fractured bedrock on steep slopes conceivably allows long groundwater flow paths to develop that will increase transit times.

The correlation between major ion concentrations and $^3$H activities may allow a first-order estimate of relative mean transit times to be made, which is useful in extending studies such as this to adjacent catchments. Caution must be exercised, however, in using the datasets as a whole as the geochemistry




may partially reflect differences in the characteristics of the peatland and eucalypt forest catchments (specifically whether the water flows through the regolith or is contained within the peat) rather than solely the mean transit times. Nevertheless, in the catchments studied here there are broad correlations between $^3$H activities and major ion concentrations within both the peatland water and

the eucalypt forest streams, which would permit estimations of relative mean transit times.

Streamflow ($Q$) is related to the mean transit time (MTT) and the volume of water held in storage ($V$) via $Q = V$ / MTT (Maloszewski and Zuber, 1982). This relationship is commonly used to estimate $V$ where MTT has been calculated and $Q$ measured (e.g., Morgenstern et al., 2010). However, in ungauged peatland catchments such as these it is possible to estimate $Q$ if the volume of peat can be

estimated and MTT is known. The area drained by the stream at Falls Rocky A is $3.5 \times 10^6$ m$^2$ and for a peat thickness of 0.5 to 1 m (which is consistent with observations at this site), $V = 1.75$ to $3.5 \times 10^6$ m$^3$. Adopting a range of mean transit times of 0.5 to 2.5 years (Table 2) yields estimates of $Q$ of $7.0 \times 10^5$ to $7.0 \times 10^6$ m$^3$ yr$^{-1}$, which span the measured average discharge of $5.64 \times 10^6$ m$^3$ yr$^{-1}$. Presuming that the parameterisation is appropriate, the correspondence between calculated and observed streamflows

occurs when mean transit times that are <1 year, which suggests that the $^3$H activities of rainfall which recharges the peat are not significantly higher than those discussed above. Carrying out these calculations on the Victorian peatland water is difficult as $Q$ tends to $\infty$ as the MTT approaches 0. In ungauged peatland catchments where mean transit times are longer, however, this would be a viable method of estimating $Q$.

**6. Conclusions**

The mean transit times in peatland water in southeast Australia are less than 6.5 years and in many cases substantially shorter. The short mean transit times implies that the peat is susceptible to drying if there are successive years of below average rainfall. In turn this makes the peat vulnerable to damage during the periodic bushfires that also occur during the summers in drought periods. The

majority of southeast Australia underwent a major drought (the "millennium drought") between 1995 and 2010 (Bureau of Meteorology, 2016) during which time there were major bushfires in the




Australian Alps, especially in 2003 and 2009. Although mainly affecting the eucalypt forests, locally these bushfires also impacted the peatlands (Arthur Rylah Institute, 2016).

The observation that the water from within the peat yields similar mean transit times to that draining the peat implies that there is not likely to be older water stored within the deeper layers. This may be

because the peat in the Victorian Alps is shallow and consequently stores little water. The peatlands do not act as a long-lived store of water to the river systems. Given their small total area, there would be little impact of peatland drying to the hydrology of the lower reaches of the adjacent catchments. However, drying of the peat could impact the hydrology of the small upland streams that form an integral part of the landscape and ecosystems of the alpine areas. The observation that even in small

eucalypt forest catchments the stream water has mean transit times of several years to decades indicates that they are likely to be more resilient to short-term variations in rainfall. As a whole, the river catchments in the Victorian Alps are eucalypt dominated, which implies that the river systems are likely to be sustained by baseflow even during prolonged drought periods, and indeed many of these streams continued to flow through the millennium drought.

More generally this study illustrates that the combination of $^3$H and major ion geochemistry allows both the timescales of water movement and the hydrogeochemical processes in these catchments to be understood. Despite the general ecological importance of peatlands, there have been few studies of water transit times. As noted by Western et al. (2009), there is a common perception that, due to their high water contents and organic rich soils, peatlands represent long-term water stores and

indeed this was the premise that this study set out to test. However, this is not the case in this study and possibly not elsewhere (e.g. the Plynlimon, Feshie, Mharcaidh and Girnock catchments that include peat areas and where transit times are <5 years: Kirchner et al., 2012; Soulsby et al., 2006; Benettin et al., 2015). Resolving mean transit times of peatland water is critical for understanding their hydrology as well as documenting their vulnerability to drying and bushfires. As the bomb pulse $^3$H

peak declines in the northern hemisphere, comparable studies such as this will become possible there which will allow a fuller assessment of peatland hydrology to be made.



*Author contributions.* U. Morgenstern was responsible for the [3]H analyses. I. Cartwright undertook the

sampling program and oversaw the analysis of the other geochemical parameters. I. Cartwright and

U. Morgenstern prepared the manuscript.

*Acknowledgements.* Funding for this project was provided by Monash University and the National

Centre for Groundwater Research and Training. The National Centre for Groundwater Research and

Training is an Australian Government initiative supported by the Australian Research Council and the

National Water Commission via Special Research Initiative SR0800001. Massimo Raveggi and Rachael

Pearson carried out the geochemical analyses at Monash University.

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





**Figure Captions**

**Fig. 1.** Location of study sites in northeast Victoria superimposed on the digital elevation model. Places: Br = Bright; Da = Dargo; MB = Mount Buffalo; MtB = Mount Beauty; MH = Mount Hotham; My = Myrtleford. FFA and FFB are the Falls Forest sampling sites A & B, other sampling sites are in the indicated study areas. Catchment boundaries from Department of Environment, Land, Water and Planning (2016)

**Fig. 2a.** Streamflow at Watchbed Creek which drains peatlands at Falls Creek (Fig. 1) between 1984 and 1987. **2b**. Flow duration curves for Watchbed Creek and Simmons Creek, which drains eucalyptus forest in the Upper Ovens (Fig. 1). Data from Department of Environment, Land, Water and Planning (2016).

**Fig. 3.** $\delta^{18}O$ vs $\delta^{2}H$ values of waters from the Australian Alps (data from Table 1) relative to the global meteoric water line (GMWL: Clark and Fritz, 1997) and the Melbourne meteoric water line (MMWL: Hughes and Crawford, 2012). Error bars are analytical uncertainties (0.2‰ for $\delta^{18}O$ and 1‰ for $\delta^{2}H$). Rainfall samples: B = Mount Buffalo multi-month aggregated samples, W = 2 week winter precipitation sample, O = other aggregated samples from southeast Australia. Linear regression line is fit to all the data.

**Fig. 4.** $^{3}H$ activities vs $\delta^{2}H$ values of waters from the Australian Alps (data from Table 1). Error bars are analytical uncertainties (1‰ for $\delta^{2}H$ and $^{3}H$ from Table 1). Rainfall samples: B = Mount Buffalo multi-month aggregated samples, W = 2 week winter precipitation sample, O = other aggregated samples from southeast Australia. Linear regression line is fit to the multi-month Mount Buffalo aggregated samples. Arrowed lines show changes in $^{3}H$ activities and $\delta^{2}H$ values due to evaporation (Evap) and radioactive decay of $^{3}H$.

**Fig. 5.** $^{3}H$ activities vs. Cl (**5a**), Na (**5b**), Si (**5e**) concentrations and Cl/Br (**5c**), Na/Cl (**5d**), Si/Cl (**5f**) ratios (data from Table 1). Arrowed lines illustrate trends expected from major geochemical processes. Ovens data are from the upper Ovens Valley streams in December 2013 and February 2014 (Cartwright



and Morgenstern, 2015); the geochemistry of these larger streams is similar to the eucalypt forest streams in the Victorian Alps.

**Fig. 6a.** Comparison of mean transit times from different lumped parameter models, calculated using Eqs 1 to 3 assuming a $^3$H activity of recharge of 3.0 TU (based on the aggregated rainfall samples). DM

5    = Dispersion model with $D_p$ of 0.1 and 1.0; EM = Exponential flow model; EPF = Exponential-piston flow model with EPM ratio = 0.3 and 1.0. **6b**. Mean transit times from the exponential flow model (EPM = 0.3) with $^3$H activities of modern recharge ranging from 3.0 to 3.4 TU. Shaded fields show $^3$H activities in the peatlands waters and eucalyptus forest streams, arrow is the median $^3$H activity in the peatland waters (data from Table 1).

10    **Fig. 7.** $^3$H activities vs. catchment areas for the peatlands and eucalyptus forest catchments. Area error bars assume a ±10% measurement error, $^3$H error bars are from the analytical uncertainties. Data from Table 1.


**Table 1.** Geochemistry of water samples from the Australian Alps

| Site[a] | Date | Area[b] | $^3$H | $\delta^2$H | $\delta^{18}$O | EC | Na | Mg | K | Ca | Si | Cl | Br | NO$_3$ | SO$_4$ |
|---|---|---|---|---|---|---|---|---|---|---|---|---|---|---|---|
| | | km$^2$ | TU | ‰ V-SMOW | ‰ V-SMOW | µS cm$^{-1}$ | mg l$^{-1}$ | mg l$^{-1}$ | mg l$^{-1}$ | mg l$^{-1}$ | mg l$^{-1}$ | mg l$^{-1}$ | mg l$^{-1}$ | mg l$^{-1}$ | mg l$^{-1}$ |
| **Peatlands** | | | | | | | | | | | | | | | |
| *Mount Buffalo* | | | | | | | | | | | | | | | |
| Buffalo Cresta A | 26/11/2015 | 1.5 | 3.32±0.053 | -26 | -5.0 | 19.12 | 1.12 | 0.125 | 0.192 | 0.621 | 2.714 | 0.94 | 0.0031 | 0.040 | 0.072 |
| Buffalo Cresta A | 23/02/2015 | 1.5 | 2.84±0.058 | -41 | -7.7 | 13.72 | 2.06 | 0.125 | 0.168 | 0.645 | 2.831 | 0.95 | 0.0039 | 0.048 | 0.185 |
| Buffalo Cresta B(P)[b] | 23/02/2015 | | 2.75±0.055 | -41 | -7.7 | 22.33 | 1.96 | 0.152 | 0.273 | 1.123 | 3.244 | 1.08 | 0.0037 | 0.285 | 0.304 |
| Buffalo Cresta C | 23/02/2015 | 2.1 | 2.75±0.056 | -40 | -7.5 | 19.92 | 1.06 | 0.121 | 0.160 | 0.432 | 1.632 | 0.95 | 0.0036 | 0.063 | 0.080 |
| Buffalo Cresta D(P) | 26/11/2015 | | 2.70±0.048 | -37 | -7.3 | 15.21 | 1.79 | 0.131 | 0.184 | 0.653 | 2.601 | 1.14 | 0.0038 | 0.073 | 0.194 |
| Buffalo Crystal A | 23/02/2015 | 3.5 | 2.74±0.055 | -35 | -6.6 | 12.01 | 1.97 | 0.153 | 0.178 | 0.684 | 2.031 | 1.01 | 0.0038 | 0.066 | 0.207 |
| Buffalo Crystal B | 23/02/2015 | 0.2 | 2.93±0.058 | -30 | -5.7 | 12.65 | 2.06 | 0.130 | 0.250 | 0.354 | 3.466 | 0.99 | 0.0037 | 0.320 | 0.245 |
| *Falls Creek* | | | | | | | | | | | | | | | |
| Falls Cope | 24/02/2015 | 1.9 | 3.14±0.061 | -41 | -7.2 | 9.62 | 0.92 | 0.247 | 0.119 | 0.519 | 1.417 | 0.47 | 0.0031 | 0.084 | 0.076 |
| Falls Langford A | 24/02/2015 | 0.5 | 3.10±0.060 | -39 | -7.1 | 21.42 | 1.43 | 0.512 | 0.203 | 0.882 | 2.496 | 0.64 | 0.0035 | 0.250 | 0.271 |
| Falls Langford B(P) | 24/02/2015 | | 2.98±0.059 | -43 | -8.3 | 8.52 | 0.88 | 0.223 | 0.160 | 0.490 | 0.812 | 0.70 | 0.0033 | 0.165 | 0.167 |
| Falls Rocky A | 24/02/2015 | 3.5 | 2.96±0.059 | -42 | -8.0 | 7.65 | 0.79 | 0.194 | 0.057 | 0.463 | 1.097 | 0.55 | 0.0026 | 0.307 | 0.111 |
| Falls Rocky A | 25/11/2015 | 3.5 | 2.90±0.050 | -42 | -8.1 | 11.25 | 0.78 | 0.149 | 0.075 | 0.318 | 2.103 | 0.51 | 0.0018 | 0.164 | 0.163 |
| Falls Rocky B(P) | 25/11/2015 | | 2.86±0.049 | -42 | -8.2 | 7.47 | 0.62 | 0.136 | 0.118 | 0.255 | 1.029 | 0.54 | 0.0023 | 0.035 | 0.130 |
| Falls Rocky C(P) | 25/11/2015 | | 3.17±0.055 | -35 | -6.5 | 9.54 | 0.79 | 0.257 | 0.245 | 0.540 | 1.125 | 0.51 | 0.0025 | 2.097 | 0.185 |
| Falls Rocky D | 25/11/2015 | 2.6 | 3.14±0.059 | -33 | -6.5 | 16.47 | 1.04 | 0.227 | 0.223 | 0.403 | 1.113 | 0.93 | 0.0033 | 0.087 | 0.079 |
| *Dargo* | | | | | | | | | | | | | | | |
| Dargo A | 25/11/2015 | 5.5 | 2.84±0.048 | -40 | -7.7 | 21.17 | 1.15 | 0.930 | 0.133 | 1.427 | 3.259 | 1.10 | 0.0040 | 0.804 | 0.142 |
| Dargo B | 25/11/2015 | 1.2 | 2.69±0.043 | -43 | -8.1 | 31.9 | 1.34 | 1.381 | 0.334 | 2.112 | 2.147 | 0.91 | 0.0038 | 0.493 | 0.050 |
| Dargo C(P) | 25/11/2015 | | 2.47±0.040 | -39 | -7.0 | 29.2 | 1.56 | 1.667 | 0.308 | 1.900 | 3.012 | 0.93 | 0.0038 | 0.388 | 0.063 |
| Dargo D | 25/11/2015 | 2.7 | 3.36±0.053 | -35 | -6.9 | 13.9 | 0.77 | 0.343 | 0.097 | 1.095 | 2.788 | 0.83 | 0.0025 | 1.845 | 0.192 |




| | | | | | | | | | | | | | | | |
|---|---|---|---|---|---|---|---|---|---|---|---|---|---|---|---|
| **Eucalypt Forest** | | | | | | | | | | | | | | | |
| Buffalo Forest A | 24/02/2015 | 3.3 | 2.21±0.048 | -35 | -6.3 | 42.7 | 3.89 | 2.056 | 0.618 | 1.727 | 4.433 | 1.16 | 0.0045 | 0.052 | 0.740 |
| Buffalo Forest A | 26/11/2015 | 3.3 | 2.35±0.042 | -34 | -6.4 | 39.5 | 3.27 | 1.878 | 0.470 | 1.536 | 4.844 | 1.30 | 0.0048 | 0.038 | 0.763 |
| Falls Forest A | 24/02/2015 | 3.3 | 1.56±0.037 | -38 | -7.4 | 65.2 | 5.70 | 3.230 | 0.543 | 3.686 | 5.060 | 2.60 | 0.0097 | 0.104 | 0.499 |
| Falls Forest B | 24/02/2015 | 1.5 | 2.10±0.047 | -37 | -7.5 | 45.7 | 3.24 | 2.092 | 0.445 | 2.149 | 4.233 | 1.09 | 0.0045 | 0.038 | 0.608 |
| Falls Forest C | 25/11/2015 | 6.4 | 2.26±0.040 | -31 | -6.2 | 45.3 | 3.41 | 1.855 | 0.566 | 1.623 | 6.236 | 1.20 | 0.0048 | 0.418 | 0.901 |
| Falls Forest D | 25/11/2015 | 5.8 | 1.70±0.035 | -37 | -7.3 | 56.0 | 4.78 | 2.793 | 0.400 | 3.092 | 9.540 | 1.22 | 0.0045 | 0.022 | 0.581 |
| Dargo Forest A | 25/11/2015 | 6.1 | 2.00±0.036 | -29 | -5.7 | 44.1 | 4.12 | 1.898 | 0.542 | 1.654 | 5.328 | 1.24 | 0.0045 | 0.240 | 0.408 |
| | | | | | | | | | | | | | | | |
| **Rainfall** | | | | | | | | | | | | | | | |
| Buffalo | Feb 2015 (12)[d] | | 2.85±0.057 | -36 | -7.0 | 11.58 | 0.68 | 0.011 | 0.058 | 0.214 | 0.081 | 0.98 | 0.0039 | 0.015 | 0.081 |
| Buffalo | Nov 2015 (9) | | 2.88±0.051 | -35 | -6.5 | 12.20 | 0.53 | 0.117 | 0.108 | 0.361 | 0.077 | 0.80 | 0.0028 | 0.106 | 0.171 |
| Buffalo | Dec 2013 (17) | | 2.99±0.046 | -38 | -7.3 | | 0.87 | | | | | 1.10 | | | |
| Buffalo | Feb 2014 (2) | | 2.90±0.049 | -37 | -7.1 | | | | | | | | | | |
| Buffalo | Jul 2014 (<1) | | 2.52±0.043 | -40 | -7.9 | | | | | | | | | | |
| Buffalo | Sep 2014 (3) | | 2.71±0.044 | -34 | -6.8 | | | | | | | | | | |
| Melbourne[e] | Jul 2013 (12) | | 2.72±0.045 | -33 | -6.3 | | | | | | | | | | |
| Otways[e] | Sep 2014 (6) | | 2.45±0.041 | -22 | -4.4 | | | | | | | | | | |
| Latrobe[e] | Aug 2015 (12) | | 2.76±0.043 | -36 | -6.8 | | | | | | | | | | |

Notes: [a] Sampling regions on Fig. 1; [b] Catchment area; [c] P = Piezometer sample; [d] Month collected and number of months sampled; [e] Other rainfall from SE Australia



**Table 2.** Calculated mean transit times

| Site[a] | Date | Mean Transit Time (yrs) | | |
|---|---|---|---|---|
| | | EPM (3.4)[b] | EPM (3.2) | EPM (3.0) |
| ***Peatlands*** | | | | |
| *Mount Buffalo* | | | | |
| Buffalo Cresta A | 26/11/2015 | <1 | ND[c] | ND |
| Buffalo Cresta A | 23/02/2015 | 3.4 | 2.2 | 1.0 |
| Buffalo Cresta B(P) | 23/02/2015 | 4.1 | 2.9 | 1.6 |
| Buffalo Cresta C | 23/02/2015 | 4.1 | 2.9 | 1.6 |
| Buffalo Cresta D(P) | 26/11/2015 | 4.4 | 3.2 | 1.9 |
| Buffalo Crystal A | 23/02/2015 | 4.2 | 3.0 | 1.7 |
| Buffalo Crystal B | 23/02/2015 | 2.8 | 1.6 | <1 |
| | | | | |
| *Falls Creek* | | | | |
| Falls Cope | 24/02/2015 | 1.4 | <1 | ND |
| Falls Langford A | 24/02/2015 | 1.7 | <1 | ND |
| Falls Langford B(P) | 24/02/2015 | 2.4 | 1.3 | <1 |
| Falls Rocky A | 24/02/2015 | 2.6 | 1.4 | <1 |
| Falls Rocky A | 25/11/2015 | 3.0 | 1.8 | <1 |
| Falls Rocky B(P) | 25/11/2015 | 3.3 | 2.1 | <1 |
| Falls Rocky C(P) | 25/11/2015 | 1.3 | <1 | ND |
| Falls Rocky D | 25/11/2015 | 1.4 | <1 | ND |
| | | | | |
| *Dargo* | | | | |
| Dargo A | 25/11/2015 | 3.4 | 2.2 | 1.0 |
| Dargo B | 25/11/2015 | 4.5 | 3.3 | 2.0 |
| Dargo C(P) | 25/11/2015 | 6.5 | 5.4 | 4.0 |
| Dargo D | 25/11/2015 | <1 | ND | ND |



| *Eucalypt Forest* | | | | |
|---|---|---|---|---|
| Falls Forest A | 24/02/2015 | 28.9 | 28.8 | 28.6 |
| Falls Forest B | 24/02/2015 | 11.5 | 10.8 | 9.6 |
| Buffalo Forest A | 24/02/2015 | 9.7 | 8.8 | 7.5 |
| Falls Forest C | 25/11/2015 | 9.1 | 8.1 | 6.6 |
| Falls Forest D | 25/11/2015 | 22.6 | 22.2 | 21.6 |
| Dargo Forest A | 25/11/2015 | 14.0 | 12.4 | 11.2 |
| Buffalo Forest A | 26/11/2015 | 7.8 | 6.9 | 5.3 |

a Sites on Fig. 1; b Mean transit times calculated using the exponential piston flow model with $^3$H activity of modern recharge given in brackets; c Not determined as $^3$H activity higher than assumed value of recharge




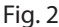


Fig. 2





Fig. 3

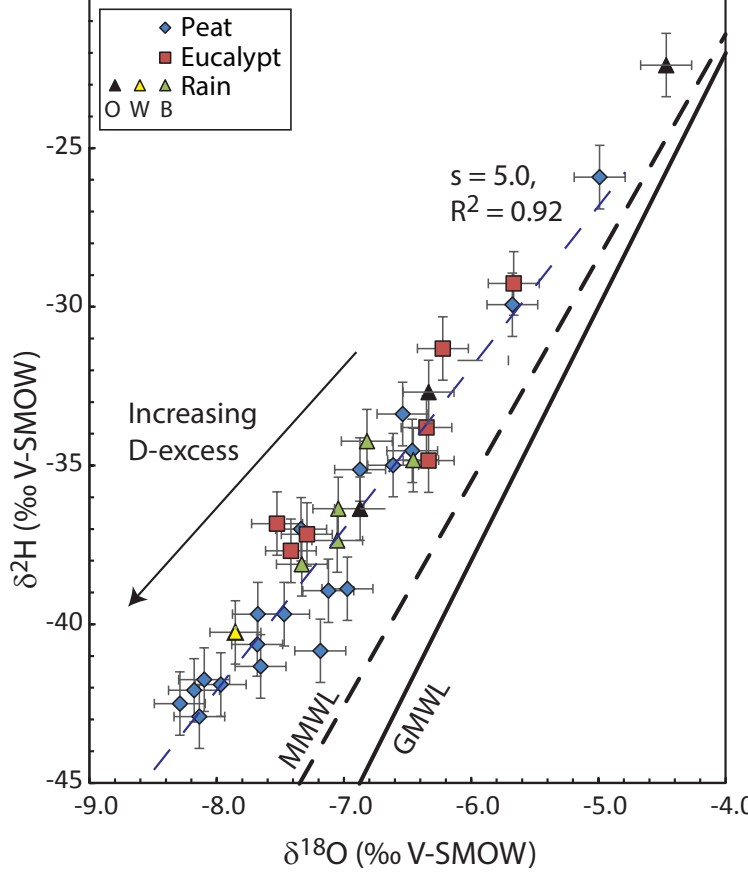





Fig. 4

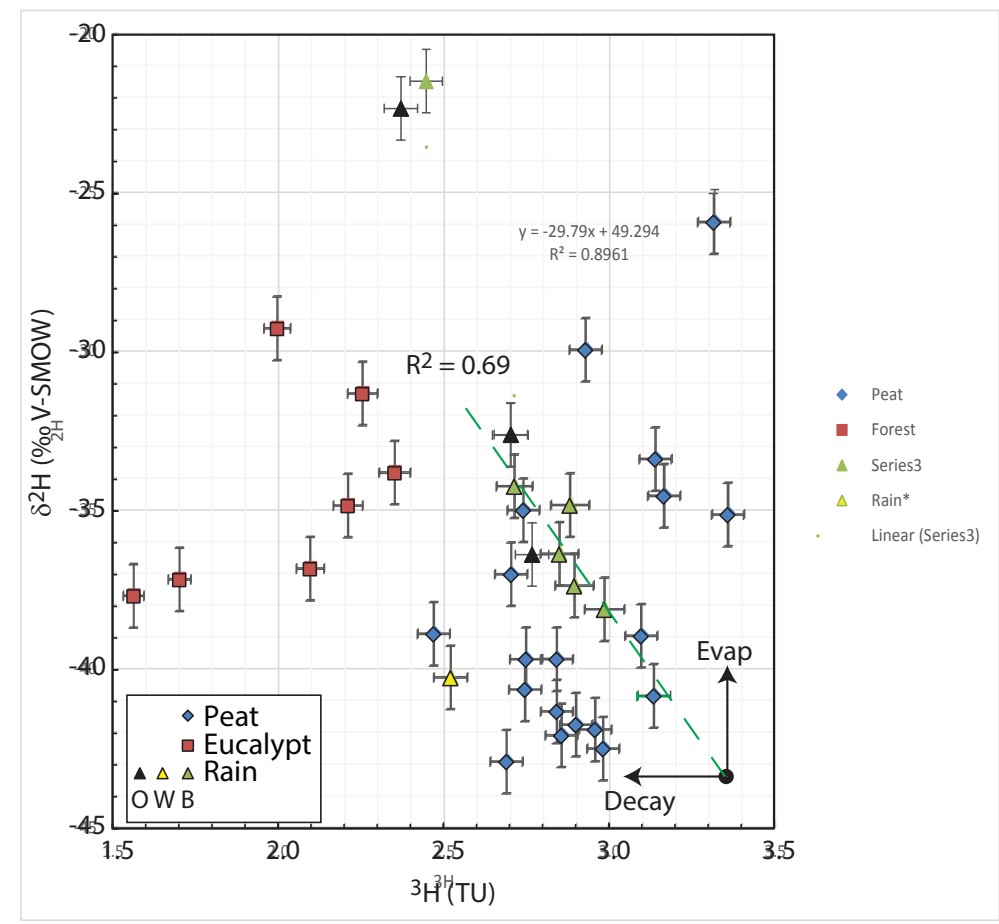





Fig. 5

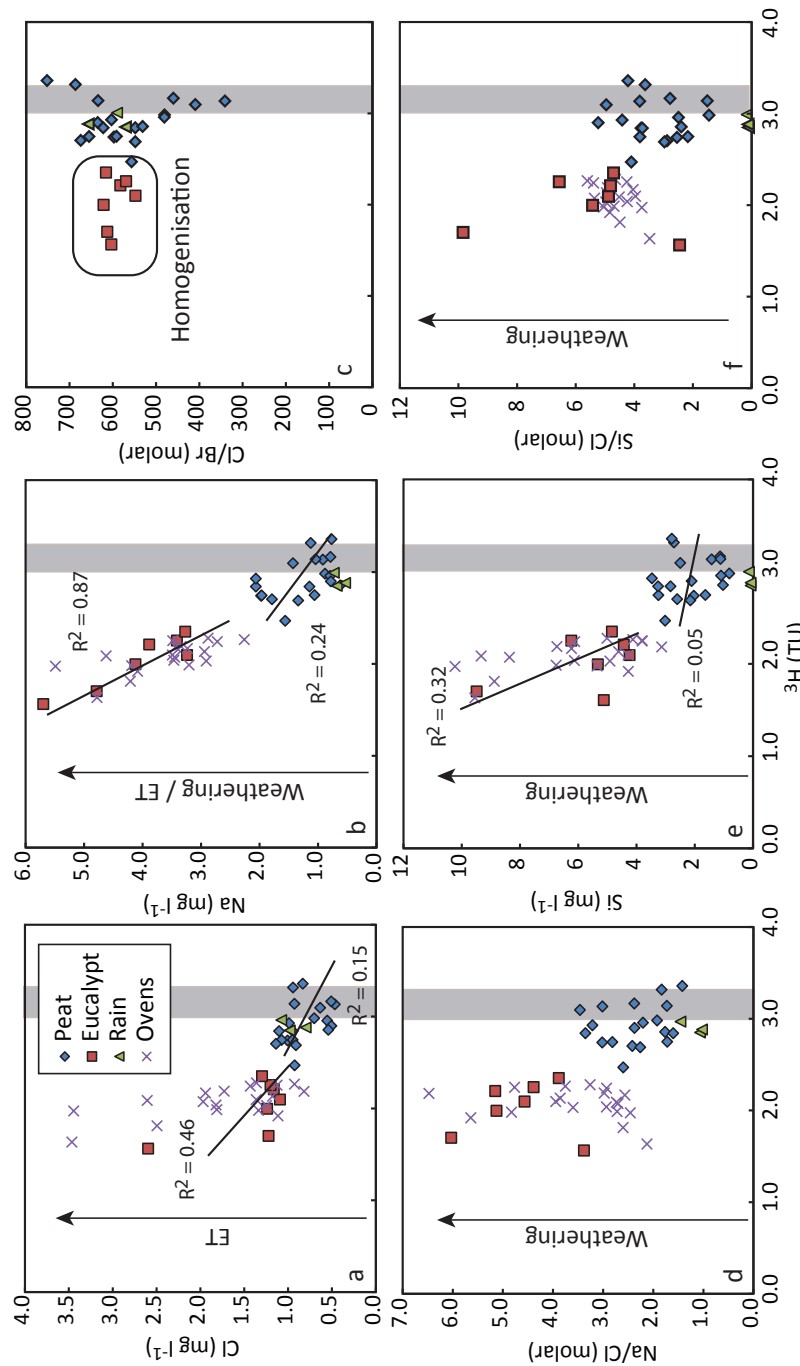





Fig. 6



Fig. 7

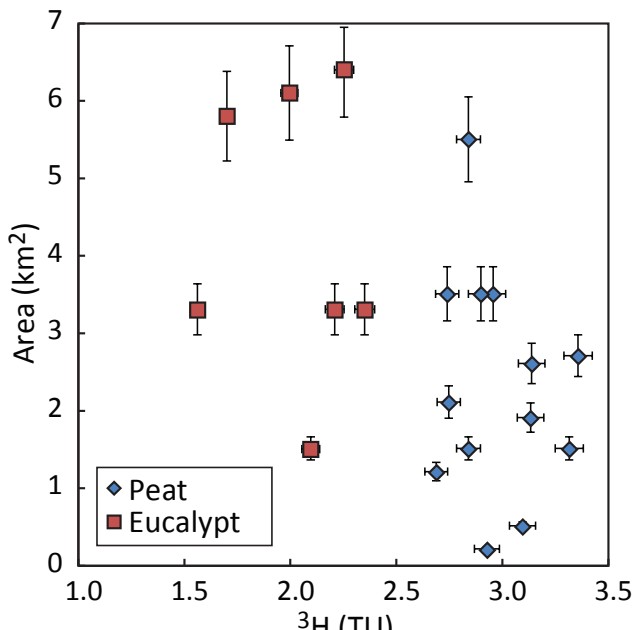