# Peer review of "Contrasting transit times of water from peatlands and eucalypt forests in the Australian Alps determined by tritium: implications for vulnerability and the source of water in upland catchments"

_Hydrology and Earth System Sciences, 2016_

## Referee Comment (RC1) · E. Timbe (Referee) · 26 Aug 2016

General Comment

The manuscript HESSD "Contrasting transit times of water from peatlands and euca-lypt forests in the Australian Alps determined by tritium: implications for vulnerability and the source of water in upland catchments" by Cartwright and Morgenstern, 2016, assesses, in terms of mean transit times, the hydrological differences between wetlands (peatlands) and eucalypt forest ecosystems. Although this study is mainly based on lumped parameter models for which uncertainties are commonly large, the inclusion in the analysis of diverse datasets like major chemical elements, stable and radioactive isotope data, allows the authors to crosscheck findings from different perspectives. The authors perform an appropriate analysis of the contrasting mean transit times found between both analyzed ecosystems, allowing to hypothesize about the hydrological functioning of the related aquifers and flow paths. Considering the scarce studies dealing with wetlands and more specifically, peatlands (e.g., as compared to mountain forest head water catchments), this study is very timely and therefore I recommend it for publication after some minor revisions which I detail below.

Specific questions/issues

Page 2, line 8. Define the acronym TU when first used, e.g., Tritium Units (TU). Besides, consider using an acronym for 3H activity/activities, this term is widely used along the manuscript (around 100 times). Please avoid beginning a sentence or a paragraph with an acronym or an abbreviation, this basic grammar rule is circumvented throughout the manuscript. Just to mention some few paragraphs starting with acronyms: pag. 8, line 17; pag. 11, lines 16 and 25, pag. 12, lines 10 and 19. Furthermore, three from four paragraphs of the Section 4.3 begin with "3H activities of..."

Page 4, lines 20 to 22. There are very few studies dealing with the appropriate tracer data resolutions to obtain reliable transit time estimations through lumped parameter models. Please consider mentioning the study by Timbe et al., 2015, who investigated this topic using stable isotopes of water.

Page 5, lines 17 to 19. Consider adding another citation, there is a more recent study for a similar ecosystem (peatlands), located in the tropics by Mosquera et al., 2016, in which mean transit times of less than one year have been also found (it uses Oxigen-18 and Deuterium as tracers).

Page 9, lines 12 to 15. Is it really possible to outline small catchment, like 0.5 km2,

from the mentioned coarse google resolution?

Technical Corrections

Page 2, line 6. Please write out the full name when the acronyms are first mentioned. E.g., define acronym 3H: "This study uses Tritium (3H) to estimate..."

Page 2, line 13. Should say: "... are higher in the eucalypt forest stream than in the peatland..."

Page 6, line 2. Rephrase: "This study is based in the upland areas of the Victorian Alps, southeast Australia and was designed to...." to something like: "This study, located in the upland areas of the Victorian Alps (southeast Australia), was designed to...".

Page 7, line 17. Please check this "the soils alci include..."?

Page 8, line 15. Please check, sentence is currently written in present tense: "Observations indicate that this is at or close..."

Page 8, line 16. Correct: "At least three bore volumes of water WERE extracted prior to sampling"

Page 8, lines 20 to 22. Rephrase the complete sentence written in these lines.

Page 9, line 3. This device is more commonly known as Thermo-Finningan Delta plus.

Page 10, line 9. Delete "transit times" once, it is repeated.

Page 11, lines 16. Please rephrase the description of the stable isotope values, it is a bit confusing in its current state. I would first describe the range for O-18 and then for Deuterium (e.g., from -8.3 to -5.0 per mil for O-18 and from -43 to -23 per mil for Deuterium).

Page 12, line 1. Rephrase "Three 2 week to 3 months aggregated samples...".

Page 12, line 4. Correct "activities in rainfall FOR this region".

Page 12, line 14. Delete the word "and" at the end of the sentence.

Page 15, line 14. Define acronym "Eqs" when first used.

Page 15, line 19. "Given that the water WAS sampled. . ." or "Given that waters WERE sampled"?

Page 16, lines 3 to 5. Rephrase the sentence contained in these lines.

Page 16, line 8. "yields mean transit times of up to 3.9 years. . ." Do you mean 4.0 years (Table 2, last column)?.

Page 17, line 1. Insert a comma after the word "increase".

Page 20, line 15. Delete the word "at" after "mean transit times".

Page 21, lines 25 – 26. Rephrase: ". . . comparable studies such as this will become possible there which will allow . . .."

Figure 1. Units are missing in the legend for elevation.

Figures 4 and 6a: Correct the axes labeling.

References

Mosquera, G. M., Segura, C., Vaché, K. B., Windhorst, D., Breuer, L. and Crespo, P.: Insights on the water mean transit time in a high-elevation tropical ecosystem, Hydrol Earth Syst Sci., 20, 2987-3004, , doi:10.5194/hess-20-2987-2016, 2016.

Timbe, E., Windhorst, D., Celleri, R., Timbe, L., Crespo, P., Frede, H.-G., Feyen, J. and Breuer, L.: Sampling frequency trade-offs in the assessment of mean transit times of tropical montane catchment waters under semi-steady-state conditions, Hydrol. Earth Syst. Sci., 19(3), 1153–1168, doi:10.5194/hess-19-1153-2015, 2015.

---

## Referee Comment (RC2) · Anonymous Referee #2 · 5 Sep 2016

The paper presents a comparison between mean transit time, MTT, between peatlands and eucalyptus forest in the Australian Alps. The authors used tritium as a tracer to model MTT using a lumped parameter approach. The authors also integrate geochemistry data and water stable isotopes in the analysis to yield important interpretations related to water storage and availability in these ecosystems. The paper is relevant to the scientific community because it provides information about underrepresented ecosystems with scare hydrologic data. The paper could eventually make an impor-

tant contribution to the scientific literature. However, I believe there are issues with the paper structure and writing style. Thus I recommend major revisions before it can be considered for publication. I provide below some specific comments and suggestions:

Abstract: Lines 1-2: the first sentence sounds redundant (use of word "that"). Please rewrite.

Line 8: define the acronym before using it (TU).

1. Introduction:

I believe the introduction should provide more information about the use of geochemical data the context of mean transit time modelling.

Page 3, line 21-23: This sentence should be written.

Page 4, section 1.1: I think the authors should include a paragraph about the recent advances and challenges in the determination of transit times and the use of lumped parameter models. For instance, the authors should clarify that the calculated times are most likely representative of base flow conditions and spell out the underlying assumptions in the use of these models. There is a vast new literature dealing with time variant modelling of transit times.

Page 4, line 19-22: there is a missing "the" before "use". In addition are there any relevant references to this statement?

Page 4 line 19: consider using "to determine" instead of "to determining"

Page 6: I suggest you eliminate section 1.3 and have the objectives of the study be the last paragraph of the introduction.

Page 6 line 11-17: this information should appear before the paragraph with the objectives. Could be part of the last paragraph in page 5.

2. Setting:

I suggest this section be part of Methods.

Page 6 line 19-22. I suggest you rewrite this sentence.

Page 7 line 1-2, line 13-14: Please use the correct notation for the scientific name of species.

3. Methods

Line 9: Explain "aggregated" over what time frame? In addition, how many, when, how frequent were the samples collected?

Line 9-10: That just means grab samples, right?

Page 9 section 3.2. More information is required about the modeling procedure, how were the best parameters identified, what objective function was used, how many possible parameter combinations were implemented, can you include dotty plots? How did you chose among the 3 different transit time functions (exponential, piston flow, and dispersion). Why did you chose these 3 and not others?

4. Results

Here is where the major structural issues arise: The results section does not present any of the MTT related findings. This is odd considering that this is precisely the main topic of the paper. Residence time results are mentioned in the abstract, discussion, and conclusions. The manuscript must include a residence time results section in which the modelling findings are presented.

Page 11 lines 2-5: This sentence is awkward. Please rewrite.

Page 11, line 8: Avoid starting the sentence with a delta symbol, instead say" Water stable isotopes ($\delta$18O and $\delta$2H).

Page 11: section 4.2. Figure 3 should be cited sooner. In addition it is not clear when or how many samples were collected.

Page 11 line 11-12: is this slope different from the GMWL. The MMWL in Fig. 3 is very step is that correct. In addition, the text says that the samples line to the left of this line which is not true.

Page 11 line 13: please provide the range if deuterium execs.

Page 11 line 22-25: This sentence is too long.

Page 12 line 5: the word "and" makes no sense.

Page 12 Lines 3-10: Did you test the data for normality to make sure a parametric test was appropriate?

Page 12 line 11-14: this sentence is too long.

Discussion:

Page 13 line 19-21: Please avoid single sentence paragraphs

Page 14 line 22: reword sentence.

Page 15: The MTT results should be move to the result section. Also the selection of the best model should be justified both physically and statistically. What objective function was use to qualify the goodness of fit?

Tables: in general the captions need to be more comprehensive. For instance, the caption for Table 1 should indicate if the isotopic values averages? If so what is the time period over which they were calculated, how many samples are included, and are there metrics of uncertainty.

Figures

Figure 1: Please use different markets to indicate the location of peatlands and eucalyptus forest.

Figure 2b. Please use probability scale in the x-axis.

Fig. 3 not clear what the dash (–) line is. The error bars for the grab samples should be representative of the accuracy (from the analysis of duplicate samples). If the precipitation signature corresponds to weighted means, then the error should be weighed errors. Please add the number of samples (n=xx) associated to each (peat, Eucalypts, rain)

Figure 4: Not sure what the equation and R2= 0.89 mean versus the R2=0.69. The legend has a "series 3" and "linear (series 3)", automatically generated from excel, that are not identified.

---

## Author Comment (AC1) · 10 Oct 2016

**Combined response to reviewers**

We thank the two referees for their helpful and insightful comments that will help improve this paper. We have addressed the various comments below (responses in blue) and have indicated how we will incorporate the suggestions in the revised paper. There are some issues that come down to editorial preferences or house style and we have indicated these in green. In addition, we understand from previous papers that a data availability section should be included after the Conclusions. We propose it to state: "All geochemistry data used in this study are contained in Table 1. Streamflow data are publicly available from the Victorian State Government Department of Environment, Land, Water, and Planning (http://data.water.vic.gov.au/monitoring.htm)."

**Reviewer 1 (Edison Timbe)**

The manuscript HESSD "Contrasting transit times of water from peatlands and eucalypt forests in the Australian Alps determined by tritium: implications for vulnerability and the source of water in upland catchments" by Cartwright and Morgenstern, 2016, assesses, in terms of mean transit times, the hydrological differences between wet-lands (peatlands) and eucalypt forest ecosystems. Although this study is mainly based on lumped parameter models for which uncertainties are commonly large, the inclusion in the analysis of diverse datasets like major chemical elements, stable and radioactive isotope data, allows the authors to crosscheck findings from different perspectives. The authors perform an appropriate analysis of the contrasting mean transit times found between both analyzed ecosystems, allowing to hypothesize about the hydrological functioning of the related aquifers and flow paths. Considering the scarce studies dealing with wetlands and more specifically, peatlands (e.g., as compared to mountain forest head water catchments), this study is very timely and therefore I recommend it for publication after some minor revisions which I detail below.

We thank the reviewer for these positive comments and while we recognise that the lumped parameter model approach is subject to uncertainty, it is probably the most practical approach to use in these situations where catchments are not intensively instrumented. In the southern hemisphere, relative mean transit times derived from $^3$H generally hold regardless of the uncertainties in the models (i.e. low $^3$H waters will be older than high $^3$H waters). This is important in this study when considering the gross differences between mean transit times in the forests and peatland. We did note this in the paper (Page 5, lines 4-9) but we can emphasise the point later as well (which in part will address comments by Reviewer 2).

Specific questions/issues

Page 2, line 8. Define the acronym TU when first used, e.g., Tritium Units (TU). Besides, consider using an acronym for 3H activity/activities, this term is widely used along the manuscript (around 100 times).

We will define "TU" on first usage both in the abstract and the main text. I tend to avoid acronyms in papers as too many make the text difficult to read, but in the case of activities, we could easily use a$^3$H without too much confusion.

Please avoid beginning a sentence or a paragraph with an acronym or an abbreviation, this basic grammar rule is circumvented throughout the manuscript. Just to mention some few paragraphs starting with acronyms: pag. 8, line 17; pag. 11, lines 16 and 25, pag. 12, lines 10 and 19.

We agree that this is poor style and will change it throughout.

Furthermore, three from four paragraphs of the Section 4.3 begin with "3H activities of…"

We will try to vary the style of sentences a little more to improve readability.

Page 4, lines 20 to 22.  There are very few studies dealing with the appropriate tracer data resolutions to obtain reliable transit time estimations through lumped parameter models. Please consider mentioning the study by Timbe et al., 2015, who investigated this topic using stable isotopes of water.

We thank the reviewer for pointing out this paper. We will incorporate it into our general discussion of lumped parameter models (Page 4, lines 19-22) for which we do not otherwise have any detail on sampling frequencies or a reference (as noted by Reviewer 2).

Page 5, lines 17 to 19. Consider adding another citation, there is a more recent study for a similar ecosystem (peatlands), located in the tropics by Mosquera et al., 2016, in which mean transit times of less than one year have been also found (it uses Oxigen-18 and Deuterium as tracers).

We became aware of the Mosquera et al. (2016) paper following the submission of our paper. As well as being one of a few studies that have addressed the mean transit times in peatlands there are results in that study (specifically the correlation of mean transit times with catchment attributes) that are also worth discussing in the context of our study and we will certainly refer to it.

Page 9, lines 12 to 15.  Is it really possible to outline small catchment, like 0.5 km2, from the mentioned coarse google resolution?

The Google Earth Pro images are high resolution and when used in conjunction with the topographic maps, the catchment areas are reasonably clearly defined. The exact catchment areas are not particularly important as there is no correlation of $^3$H activities with area (this is a recurring theme in our studies of Australian catchments). We can add a sentence noting that the catchment areas are approximate and review our error bars on Fig. 7, but this does not change any of our conclusions. We also have annotated Google images of the sampling sites that we can include as supplementary material to provide context

Technical Corrections

Page 2, line 6. Please write out the full name when the acronyms are first mentioned. E.g., define acronym 3H: "This study uses Tritium (3H) to estimate…"

Agreed (as above)

Page 2, line 13.  Should say:  "are higher in the eucalypt forest stream than in the peatland…"

Agreed, we will rephrase this sentence

Page 6, line 2. Rephrase: "This study is based in the upland areas of the Victorian Alps, southeast Australia and was designed to..." to something like: "This study, located in the upland areas of the Victorian Alps (southeast Australia), was designed to…"

This is an awkward sentence that we will rephrase

Page 7, line 17. Please check this "the soils alci include…"?

This sentence does not convey what it is supposed to and needs rewriting (the colluvium and alluvium are nor related to the soils so need splitting out).

Page 8, line 15. Please check, sentence is currently written in present tense: "Observations indicate that this is at or close…"

We will change this sentence so that the tense is consistent.

Page 8, line 16. Correct: "At least three bore volumes of water WERE extracted prior to sampling"

Agreed that it should be "were" not "was".

Page 8, lines 20 to 22. Rephrase the complete sentence written in these lines.

We agree that this is not clearly written and will rephrase it.

Page 9, line 3. This device is more commonly known as Thermo-Finningan Delta plus.

The device is as named on the manual etc.

Page 10, line 9. Delete "transit times" once, it is repeated.

We will correct this as suggested.

Page 11, lines 16. Please rephrase the description of the stable isotope values, it is a bit confusing in its current state. I would first describe the range for O-18 and then for Deuterium (e.g., from -8.3 to -5.0 per mil for O-18 and from -43 to -23 per mil for Deuterium).

Yes, the ranges should be for $\delta^{18}O$ and $\delta^2H$. We will correct this as suggested.

Page 12, line 1. Rephrase "Three 2 week to 3 months aggregated samples…"

Probably "Three aggregate samples from periods of 2 weeks to 3 months…" would be clearer and more grammatically correct.

Page 12, line 4. Correct "activities in rainfall FOR this region".

We will correct this as suggested.

Page 12, line 14. Delete the word "and" at the end of the sentence.

We will correct this as suggested.

Page 15, line 14. Define acronym "Eqs" when first used.

In other HESS papers "Eqs" is generally used without definition (i.e., it is one of the standard acronyms), but we can correct it if it fits with the house style.

Page 15, line 19. "Given that the water WAS sampled" or "Given that waters WERE sampled"?

Should be "waters were".

Page 16, lines 3 to 5. Rephrase the sentence contained in these lines.

"For a water with a $^3H$ activity of 3 TU, propagating the analytical uncertainty of… results in…" would be clearer.

Page 16, line 8. "yields mean transit times of up to 3.9 years" Do you mean 4.0 years (Table 2, last column)?.

Yes this should be 4 years. We will change this and ensure that the other values are the same as those in the table.

Page 17, line 1. Insert a comma after the word "increase".

We will correct this as suggested.

Page 20, line 15. Delete the word "at" after "mean transit times".

We will correct this as suggested.

Page 21, lines 25 – 26.  Rephrase:  "…comparable studies such as this will become possible there which will allow..."

This would be better as "As the remaining bomb-pulse $^3H$ declines in the northern hemisphere, the utility of $^3H$ in determining mean transit times will be improved, which will allow a fuller assessment…"

Figure 1. Units are missing in the legend for elevation.

We will add the units to the legend (they are m)

Figures 4 and 6a: Correct the axes labelling

For some reason hidden layers (from Adobe CS) became visible in these figures following uploading (they were not visible in the files that were uploaded or .pdf's that I generated from these on my computer). We will remove the hidden layers in the final version of the figures.

Reviewer 2

The paper presents a comparison between mean transit time, MTT, between peatlands and eucalyptus forest in the Australian Alps.  The authors used tritium as a tracer to model MTT using a lumped parameter approach. The authors also integrate geochemistry data and water stable isotopes in the analysis to yield important interpretations related to water storage and availability in these ecosystems.  The paper is relevant to the scientific community because it provides information about underrepresented ecosystems with scare hydrologic data.  The paper could eventually make an important contribution to the scientific literature. However, I believe there are issues with the paper structure and writing style.  Thus I recommend major revisions before it can be considered for publication.

We also thank this reviewer for the comments that recognise the importance of this study and the dearth of information currently available from peatlands. One of the reviewer's major comments relates to the structure of the paper. HESS does not have a fixed house style and looking through recent issues, the structure that we have used is one that is common but not ubiquitous. We discuss this in more detail below, but perhaps the editor can comment on what structure they think would best fit with the journal. Another group of comments relates to the application of the lumped parameter models and there are some important differences between the use of single $^3$H measurements in the southern hemisphere and other tracers that are applied as time series. Although we note this in section 1.2 (and it is discussed in detail elsewhere, notably Morgenstern et al., 2010), it would be worth adding more details here. While not wishing to turn this into a review, a few more details in the introduction and throughout the paper would be helpful. This would also help address the general comments of Reviewer 1.

I provide below some specific comments and suggestions:

Abstract: Lines 1-2: the first sentence sounds redundant (use of word "that"). Please rewrite.

The second "that" should be a "which"

Line 8: define the acronym before using it (TU).

Agreed we will define it on first use both in the abstract and main text (also raised by Reviewer 1)

1. Introduction:

I believe the introduction should provide more information about the use of geochemical data the context of mean transit time modelling.

While not trying to turn what is an applied study into a review article, we agree that a few more key details would be useful. We have considered the issue of aggregation and time-invariance and will add more detail on these topics in the introduction and where appropriate later in the paper (e.g., Section 5.2) (see below).

Page 3, line 21-23: This sentence should be written.

We will clarify this sentence.

Page 4, section 1.1: I think the authors should include a paragraph about the recent advances and challenges in the determination of transit times and the use of lumped parameter models. For instance, the authors should clarify that the calculated times are most likely representative of base flow conditions and spell out the underlying assumptions in the use of these models. There is a vast new literature dealing with time variant modelling of transit times.

As indicated above we are happy to do this without turning the paper into a review. The recent work by Kirchner (2016. Hydrol. Earth Syst. Sci., 20, 299–328) provides a good summary of the problem of transience for mean transit time calculations and will be used to frame this discussion. The assumption regarding time-invariance is important where mean transit times are calculated from time series measurements (which is the methodology used for stable isotope, major ion tracers, or tritium in the northern hemisphere), which was explored in depth in the Kirchner paper.

Our approach was to estimate the mean transit times from individual $^3$H measurements rather than to use a time series approach, which due to the attenuation of the bomb-pulse tritium is not now practicable for $^3$H in the southern hemisphere. The attenuation of the bomb-pulse tritium, however, now allows for robust interpretation of individual $^3$H measurements. These $^3$H activities reflect the residence time in the catchment and do not require an assumption of time invariance. What is required is that the flow path geometry remains relatively constant and able to be approximated by an LPM. The disadvantage of the individual $^3$H measurement approach is that one must assume a LPM.

By contrast, the time series approach using $^3$H, which is still possible in the northern hemisphere, does allow one to assess how appropriate the assumed LPM is. The disadvantages are that there is an assumption of time invariance between the various sampling times, which is unlikely in reality. The time series approach also only yields a single mean transit time estimate per site which would not apply to different times if the flow system changed, and therefore it is not suited for interrogating whether the mean transit times vary with flow conditions or in different years.

It was noted in this section that in the southern hemisphere it is possible to use $^3$H to provide estimates of mean transit times from single measurements (Page 5, lines 6-7) rather than having to use time series data. We will explain more fully the differences in these two approaches. This issue came up again in Section 3.2 and we will reiterate the points there also. Also, again due to the attenuation of the bomb-pulse tritium, the relative ages derived from $^3$H activities in the southern hemisphere are not model dependant (Page 5, lines 7-9) and it is worth emphasising this throughout the paper.

Aggregation (as in the mixing of waters from different flow systems with different mean transit times) is a problem for all methods that determine mean transit times (e.g., Kirchner 2016, Hydrol. Earth Syst. Sci., 20, 279–297), and we will also add a comment on that to this part of the text (and discuss the potential impacts later). A recent paper by Stewart et al. (2016, Hydrol. Earth Syst. Sci. Discussions) directly addresses aggregation for mean transit times from single $^3$H activities. That study demonstrates that for single $^3$H measurement, aggregation is most problematic when waters with very different mean transit times are mixed. The figure below shows the results of aggregating up to 5 samples with mean transit times of between 10 and 50 years in random amounts (150 simulations). The mixed mean transit time is calculated from the proportion of the end members (i.e., a 1:1 mix of waters with mean transit times of 10 and 50 years has a mixed MTT of 30 years) and the apparent MTT is calculated from the $^3$H activity of the mixed sample. All the mean transit times were calculated from the exponential-piston flow model with f = 0.75. The red line is agreement between the mixed and apparent mean transit times.

Aggregation in this case results in an underestimation of the mean transit times (which is in agreement with the Kirchner and Stewart studies). The maximum error is when there are only two end-members being mixed and there are approximately equal amounts of each end-member (the black line shows the apparent ages from binary mixes of waters with mean transit times of 10 and 50 years). Repeated aggregation reduces the error. For example, aggregating two waters with mean transit times of 10 and 50 years in a 1:1 ratio results in a difference between apparent and mixed mean transit times of ~13%; by contrast, if nine waters with MTT's of 10, 15, 20, 25, 30, 40, 45 & 50 years were aggregated in equal amounts the difference is 3.6%.

[Figure]

The percentage differences are similar for other LPM's and for end-members with different mean transit times.

Given the recent discussion of aggregation in the literature, we propose to include a statement on it in the introduction to determining mean transit times and a paragraph in section 5.2 that addresses the impacts along the lines of the above discussion (albeit shorter). Overall, the uncertainty resulting from aggregation is similar to that of the other uncertainties and does not change the overall conclusions.

The reviewer is correct that we are characterising baseflow conditions and we will note this where we outline the aims of the study (Section 1.3). This is also important for the assumption of a single water store in our calculations (Section 5.3) and again we will reiterate it there.

Page 4, line 19-22: there is a missing "the" before "use". In addition are there any relevant references to this statement?

Reviewer 1 noted that the Mosquera et al. (2016) reference has information on sampling frequency, and we can incorporate the details from that paper here.

Page 4 line 19: consider using "to determine" instead of "to determining"

Will change as suggested.

Page 6: I suggest you eliminate section 1.3 and have the objectives of the study be the last paragraph of the introduction.

We are happy to do this as it makes little difference to the paper's structure. However, previously we have been requested to provide a specific "Objectives" section in papers. Perhaps the Editor could comment on what fits better with HESS's house style.

Page 6 line 11-17: this information should appear before the paragraph with the objectives. Could be part of the last paragraph in page 5.

We agree that this doesn't fit with the objectives (whether they are in a separate section or not) and we will move it as suggested.

I suggest this section be part of Methods.

We are happy to do this although it makes little difference to the paper's overall structure. Again HESS doesn't appear to have a consistent house style. Perhaps the Editor could comment

Page 6 line 19-22. I suggest you rewrite this sentence.

Would probably be better as "Water from peatlands and eucalypt forest streams…"

Page 7 line 1-2, line 13-14:  Please use the correct notation for the scientific name of species.

We will italicise the species names

3. Methods

Line 9: Explain "aggregated" over what time frame? In addition, how many, when, how frequent were the samples collected?

"Aggregated" refers to the total rainfall collected over the time period. The duration and the frequency of rainfall samples is in Table 1 but that information can be added here so that it is clear what timespans and when the rainfall is collected.

Line 9-10: That just means grab samples, right?

We can use the term "Grab samples" if it is clearer.

Page 9 section 3.2.  More information is required about the modeling procedure, how were the best parameters identified, what objective function was used, how many possible parameter combinations were implemented, can you include dotty plots?  How did you chose among the 3 different transit time functions (exponential, piston flow, and dispersion). Why did you chose these 3 and not others?

As discussed above, to calculate mean transit times from single $^3$H measurements one has to assume a LPM model. Our choice of models spans those commonly used in the literature. For example of the ~120 studies listed by McGuire & McDonnell (Journal of Hydrology, 330, 543– 563) some 100 utilise the three LPMs that we discuss. The exponential-piston flow model corresponds to the most likely form of the flow system (near vertical recharge through the unsaturated zone and exponential flow in the unconfined shallow aquifers and soils). Where $^3$H time series data are available, this model with f values between 1.0 and 0.4 has reproduced the $^3$H activities of river water and groundwater (e.g. Maloszewski et al. 1983, J. Hydrol., 66, 319–330; Stewart et al. 2007, Hydrol. Process., 21, 3340–3356).

There are differences between the mean transit times from the different models and we have used this (recognising that we do not know which is the most appropriate LPM) to place some uncertainties on the mean transit times. Given that the mean transit times are so different between

the peatland and eucalyptus forest waters, the main conclusion of the paper (that water retention times in the peat are much shorter than those in the forests) remains unaffected by the choice of model (we note this point on Page 20, lines 12-14).

We will add more details here as to the rationale behind the modelling and our choice of models, which will echo the additional material that will appear in the introduction (discussed above). Additionally, we will re-emphasise these points in Section 5.2 where we discuss the results of the modelling.

4. Results

Here is where the major structural issues arise: The results section does not present any of the MTT related findings. This is odd considering that this is precisely the main topic of the paper. Residence time results are mentioned in the abstract, discussion, and conclusions. The manuscript must include a residence time results section in which the modelling findings are presented.

The paper is structured such that it presents the data first (Section 4) and then discusses it (Section 5). While this is not a strict requirement in HESS, it is a common structure used by many journals and one that is followed by numerous papers in HESS and elsewhere (and from experience it is a structure that some reviewers are insistent on). While it is possible to write papers that mix results and interpretation into a single section, the drawback of doing this is that the reader may be unclear as to what is data and what is interpretation or that the paper begins to present interpretations ahead of describing the data that was used to make those interpretations. Much of what precedes section 5.2 is required to produce the interpretations in that section.

We take the point that section 5.2 is a fair way into the paper, but we would prefer to keep the structure as is. What we can do is to provide a brief explanation as to what is in each section at the start of Sections 4 and 5 (i.e. that this is the section that presents the data that is interpreted in Section 5) which will make it clearer what the reader can expect.

Page 11 lines 2-5: This sentence is awkward. Please rewrite.

We can rephrase this sentence to ensure that it is clear.

Page 11, line 8: Avoid starting the sentence with a delta symbol, instead say" Water stable isotopes (18O and 2H).

A similar point was raised by Reviewer 1. We will ensure that sentences do not start with symbols or acronyms, which we agree is poor grammar.

Page 11: section 4.2. Figure 3 should be cited sooner.

We will cite this on first usage (top of Section 4.2).

In addition it is not clear when or how many samples were collected.

The samples are listed in Table 1 together with their sampling dates. We will ensure that that is clear in the figure caption / legend and in the methods.

Page 11 line 11-12: is this slope different from the GMWL. The MMWL in Fig. 3 is very step is that correct. In addition, the text says that the samples line to the left of this line which is not true.

The MMWL has a slope of ~7.5 (vs. 8 for the GMWL), which we will note in the text. The line through the samples is a best fit to the data with a slope of 5 (it is not the MMWL), we will ensure that that is clear in the figure caption and the text.

Page 11 line 13: please provide the range if deuterium execs.

We will add these to the text.

Page 11 line 22-25: This sentence is too long.

Agreed that it would be clearer if we split it.

Page 12 line 5: the word "and" makes no sense.

This is also a long sentence that we could make clearer by splitting.

Page 12 Lines 3-10: Did you test the data for normality to make sure a parametric test was appropriate?

Although there is no indication that the data are not normally distributed, the datasets are probably too small for the statistics to be rigorous. Rather than presenting the p values, we can just point out that the $^3$H activities between the sites overlap (the p values were not used for any other purpose).

Page 12 line 11-14: this sentence is too long.

We can split the sentence to make it clearer.

Discussion:

Page 13 line 19-21: Please avoid single sentence paragraphs.

The lines listed are not a single sentence paragraph. The first paragraph to section 5 is a single sentence, but it introduces the section. Given that we indicated that we would provide a bit more detail of what Sections 4 and 5 contain, that section will be a little longer.

Page 14 line 22: reword sentence.

We will clarify this sentence

Page 15: The MTT results should be move to the result section. Also the selection of the best model should be justified both physically and statistically. What objective function was use to qualify the goodness of fit?

We discussed the location of this material and the way we approached the modelling above. Our preference is to retain this material here as it constitutes interpretation (discussion) rather than data presentation (results). As noted above we will provide more detail as to the choice of models and the assumptions that we have made, and some of those details will go into the discussion of the results in this section. We start off discussing the exponential-piston flow model as this accounts for

vertical flow in the unsaturated zone and exponential flow in the saturated zone (which is how the flow system is most likely to operate). However, we do recognise that this is only one possibility, which is why we utilised the other LPM's for comparison. However, to reiterate that adopting different models does not alter the primary conclusion that the mean transit times in the peatlands must be much shorter than those in the forests.

Tables: in general the captions need to be more comprehensive. For instance, the caption for Table 1 should indicate if the isotopic values averages? If so what is the time period over which they were calculated, how many samples are included, and are there metrics of uncertainty.

These data are from single samples. The error bars on the figures reflect analytical errors not sample variability (as is stated in the figure captions). We can specify this in the Table caption (it would make it clearer for the rainfall samples that were collected over a period of time).

Figures

Figure 1: Please use different markets to indicate the location of peatlands and eucalyptus forest.

It would be difficult to show the outcrops of peat at this scale. As noted above, we have annotated high-resolution Google Earth images that show the distribution of peatlands and eucalypt forest more clearly that we could include as a supplement.

Figure 2b. Please use probability scale in the x-axis.

Although %time exceeded is commonly used, probability is more accurate (the graph remains the same).

Fig. 3 not clear what the dash (–) line is.

The dashed line is a linear best fit to the rainfall samples. We will add that to the figure caption.

The error bars for the grab samples should be representative of the accuracy (from the analysis of duplicate samples).

This is the case (as in the figure caption).

If the precipitation signature corresponds to weighted means, then the error should be weighed errors.

They are individual samples.

Please add the number of samples (n=xx) associated to each (peat, Eucalypts, rain)

We agree that this would be useful and will add it to the figure.

Figure 4: Not sure what the equation and R2= 0.89 mean versus the R2=0.69. The legend has a "series 3" and "linear (series 3)", automatically generated from excel, that are not identified.

As explained above, some hidden layers appeared in the final versions of these figures when they were uploaded. We will remove them.

---

## Author Response (AR1)

**Summary of manuscript corrections**

We have addressed the various comments in our response to the reviewers and have incorporated most the suggestions in the revised paper (as summarised in blue). Given that many of the minor comments related to sentences that were long or which contained multiple clauses, we have also done additional editing to remove some of these. Finally, we have included a data availability section after the Conclusions as was requested on a previous paper.

**Editorial Comments**

(1) while it is true that tritium does carry quite some information content about the age of water, I think the often communicated "one sample is enough" message needs to be toned down a bit (p.5,l.6). of course one sample is better than no sample, but given the observational and modelling uncertainties involved, it will only give us a very rough estimate. in principle, models could be calibrated using one O-18 value as well. the question is then, however, how much confidence do we have in this one sample and how strongly can it constrain redundant parameterizations? the same is of course true for tritium.

We didn't really intend the comment in that way. What we are trying to convey is that $^3$H can now be used in a similar way to other radioisotope tracers (e.g. $^{14}$C or $^{36}$Cl) where an age or mean transit time estimate can be derived from individual measurements. We agree that you cannot adequately characterise a catchment with one sample (with $^3$H or any of the other radioisotope tracers). However, the use of $^3$H in this way facilitates studying the catchment at different flow conditions and probably allows more sites to be studied. We have discussed this in more detail on pages 5 and 6 and tried to emphasise the differences in approaches between the time series and the single measurement approach. The time-series approach has the advantage that it can be used to determine the most appropriate LPM by comparison of observed and predicted $^3$H vs. time. However, it requires an assumption of time invariance which may not always be reasonable. The single measurement approach does not require an assumption of time invariance but it cannot be used to determine the most appropriate LPM.

In the southern hemisphere, it is questionable whether the time series approach would work for $^3$H in the future due to the much lower $^3$H activities of the bomb pulse waters, which have now all but disappeared. For example, the calculated decrease of $^3$H activities for a water with a mean transit time of 10 years between 2016 and 2026 as predicted by the EPF and EM models used in the paper with the Melbourne $^3$H record is only 0.2 TU. The corresponding decline for a northern hemisphere site (e.g. Ottawa) is around 3 TU. Additionally, the time vs. $^3$H trends of the different models in the southern hemisphere are similar within analytical uncertainty.

Both approaches have advantages and drawbacks and there is no "best" method. The use of $^3$H is obviously in a transitional phase as the bomb pulse disappears from requiring a time-series approach to being used as a conventional age tracer. Hopefully, we have conveyed that in the paper.

(2) in your replies you mention that MTT studies in peatlands are rare. this is not exactly true. most of the british MTT work (e.g. soulsby, tetzlaff, birkel and myself) is located in peatlands.

We did cite a number of these but have added a couple more. If one does a survey of the literature the number of studies specifically on peatland catchments is much less than on other landscapes, but there clearly are some. We have rephrased the start of this section to reflect that (page 6, lines 18-19).

(3) in one comment reviewer 2 highlighted the missing description of the model calibration strategy - i.e. how were the parameters determined? what were the objective functions used? what was the calibrated model performance? you did not address this point in your replies: you only mention your assumptions on the choice of the model but not on how you obtained the parameters. please add that in the revised manuscript.

As we noted in the response to the reviewers, it is not possible to calibrate the model using the $^3$H data. Using $^3$H as a "conventional" radioisotope tracer requires one to assign a plausible lumped parameter model (as is the case for other commonly-used tracers such as $^{14}$C in older waters). We have approached the calculation of mean transit times by utilising a range of lumped parameter models and considering the different results as part of the uncertainties in the transit time calculations. This goes back to the point regarding the differences between the use of single $^3$H activities in the southern hemisphere and the need/ability to use time series $^3$H data in the northern hemisphere. Our choice of models was based on those most commonly used for determining mean transit times (e.g. as reviewed by McGuire and McDonnell, 2006). LPMs such as the exponential-piston flow model correspond to the likely geometry of the flow system and is the preferred model. We have tried to be more explicit on this point in Sections 1.1 (pages 5 & 6), 2.3 (page 11), 4.2 (page 17-19) and 5 (page 24). We also added the mean transit times from the other LPM's to Table 2 and discuss them explicitly in the paper (page 17).

(4) p.6,l.11-17 would indeed better fit into the methods section as suggested by reviewer 2

We made this change

(5) apart from that i think you can leave the introduction section and its sub-sections structurally as is.

We left the other sections as in the original manuscript.

**Reviewer 1 (Edison Timbe)**

The manuscript HESSD "Contrasting transit times of water from peatlands and eucalypt forests in the Australian Alps determined by tritium: implications for vulnerability and the source of water in upland catchments" by Cartwright and Morgenstern, 2016, assesses, in terms of mean transit times, the hydrological differences between wet-lands (peatlands) and eucalypt forest ecosystems. Although this study is mainly based on lumped parameter models for which uncertainties are commonly large, the inclusion in the analysis of diverse datasets like major chemical elements, stable and radioactive isotope data, allows the authors to crosscheck findings from different perspectives. The authors perform an appropriate analysis of the contrasting mean transit times found between both analyzed ecosystems, allowing to hypothesize about the hydrological functioning of the related aquifers and flow paths. Considering the scarce studies dealing with wetlands and more specifically, peatlands (e.g., as compared to mountain forest head water catchments), this study is very timely and therefore I recommend it for publication after some minor revisions which I detail below.

In response to this comment and the more extensive comments from the second reviewer regarding lumped parameter modelling and uncertainties. This additional discussion now appears in Section 4.2 (page 17, lines 4-7 and page 19, lines 12-14).

Specific questions/issues

Page 2, line 8. Define the acronym TU when first used, e.g., Tritium Units (TU). Besides, consider using an acronym for 3H activity/activities, this term is widely used along the manuscript (around 100 times).

We defined "TU" on first usage both in the abstract (page 2, line 7) and the main text (page 4, line 11). We did not abbreviate activities as it is generally not done so for $^3H$.

Please avoid beginning a sentence or a paragraph with an acronym or an abbreviation, this basic grammar rule is circumvented throughout the manuscript. Just to mention some few paragraphs starting with acronyms: pag. 8, line 17; pag. 11, lines 16 and 25, pag. 12, lines 10 and 19.

We changed these occurrences and others throughout the paper

Furthermore, three from four paragraphs of the Section 4.3 begin with "3H activities of…"

We varied the style of sentences to improve readability (now section 3.3, pages 13-14).

Page 4, lines 20 to 22. There are very few studies dealing with the appropriate tracer data resolutions to obtain reliable transit time estimations through lumped parameter models. Please consider mentioning the study by Timbe et al., 2015, who investigated this topic using stable isotopes of water.

We thank the reviewer for pointing out this paper which we have referenced (page 4, line 23).

Page 5, lines 17 to 19. Consider adding another citation, there is a more recent study for a similar ecosystem (peatlands), located in the tropics by Mosquera et al., 2016, in which mean transit times of less than one year have been also found (it uses Oxigen-18 and Deuterium as tracers).

We became aware of the Mosquera et al. (2016) paper following the submission of our paper and have now referenced it in the discussion on page 6, line 28.

Page 9, lines 12 to 15. Is it really possible to outline small catchment, like 0.5 km2, from the mentioned coarse google resolution?

The Google Earth Pro images are high resolution and when used in conjunction with the topographic maps, the catchment areas are clearly defined. The exact catchment areas are not particularly important as there is no correlation of $^3H$ activities with area (this is a recurring theme in our studies of Australian catchments).

We have added the detailed catchment maps as a supplementary Figure.

Technical Corrections

Page 2, line 6. Please write out the full name when the acronyms are first mentioned. E.g., define acronym 3H: "This study uses Tritium (3H) to estimate…"

This was done on first use in the abstract (page 2, line 6) and main text (page 4, line 11).

Page 2, line 13. Should say: "are higher in the eucalypt forest stream than in the peatland…"

We rephrased this sentence (page 2, lines 13-14)

Page 6, line 2. Rephrase: "This study is based in the upland areas of the Victorian Alps, southeast Australia and was designed to…" to something like: "This study, located in the upland areas of the Victorian Alps (southeast Australia), was designed to…"

We rephrased this sentence to "This study was designed to test the hypothesis that peatlands in the Victorian Alps, southeast Australia represent a relatively long-lived store of water" which more clearly conveys the aims of the study (page 7, lines 11-12).

Page 7, line 17. Please check this "the soils alci include…"?

We split this material up and corrected the typo "alci". It now reads "These soils overlie weathered regolith that is a few tens-of-centimetres to a few metres thick. There are also minor deposits of colluvium and alluvium along the streams (Shugg, 1987). Groundwater flow is restricted to the weathered zones and fractures in the granites and metasediments; the minor alluvial and colluvial sediments are more permeable but represent only a minor part of the landscape" which is clearer (page 8, lines 23-27)

Page 8, line 15. Please check, sentence is currently written in present tense: "Observations indicate that this is at or close…"

This is now in the past tense to be consistent with the rest of the paragraph (page 9, line 27)

Page 8, line 16. Correct: "At least three bore volumes of water WERE extracted prior to sampling"

This was corrected as suggested (page 10, lines 1-2).

Page 8, lines 20 to 22. Rephrase the complete sentence written in these lines.

We rephrased this as "Enrichment of $^{3}$H was by a factor of 95, which results in a detection limit of 0.02 TU, while the use of deuterium-calibration for each sample results in a 1% reproducibility of the tritium enrichment." (page 10, lines 6-8).

Page 9, line 3. This device is more commonly known as Thermo-Finningan Delta plus.

The device is as named on the manual etc.

Page 10, line 9. Delete "transit times" once, it is repeated.

This was corrected (page 11, line 22).

Page 11, lines 16. Please rephrase the description of the stable isotope values, it is a bit confusing in its current state. I would first describe the range for O-18 and then for Deuterium (e.g., from -8.3 to -5.0 per mil for O-18 and from -43 to -23 per mil for Deuterium).

We corrected this as suggested (page 13, line 11).

Page 12, line 1. Rephrase "Three 2 week to 3 months aggregated samples…"

To also address comments by reviewer 2, we have changed this to "The $^3$H activities of three multi-month rainfall samples from Mount Buffalo are 2.85 TU (12 month sample collected in February 2015), 2.88 TU (9 month sample collected in November 2015), and 2.99 (17 month sample collected in December 2013) (Table 1). Three samples of rainfall collected over periods of 2 weeks to 3 months in 2014 also from Mount Buffalo have $^3$H activities of 2.52 to 2.90 TU (Table 1)." (page 13, lines 21-25).

The word "aggregated" was probably not clear and is also used to describe mixing of the waters in the catchment, so we have removed it when referring to the rainfall.

Page 12, line 4. Correct "activities in rainfall FOR this region".

We corrected this as suggested (page 13, line 27).

Page 12, line 14. Delete the word "and" at the end of the sentence.

This was corrected (Page 14, line 10).

Page 15, line 14. Define acronym "Eqs" when first used.

In other HESS papers "Eqs" is generally used without definition (i.e., it is one of the standard acronyms). We left this as is.

Page 15, line 19. "Given that the water WAS sampled" or "Given that waters WERE sampled"?

Changed to "waters were" (page 17, line 14).

Page 16, lines 3 to 5. Rephrase the sentence contained in these lines.

Changed to "For a water with a $^3$H activity of 3 TU, propagating the analytical uncertainty of ±2% produces an uncertainty in mean transit times of ±0.3 years" (page 18, lines 8-9).

Page 16, line 8. "yields mean transit times of up to 3.9 years" Do you mean 4.0 years (Table 2, last column)?.

Changed to 4.0 years (page 18, line 14).

Page 17, line 1. Insert a comma after the word "increase".

Corrected as suggested (page 18, line 18)

Page 20, line 15. Delete the word "at" after "mean transit times".

Corrected as suggested (page 23, line 9)

Page 21, lines 25 – 26.  Rephrase:  "…comparable studies such as this will become possible there which will allow..."

We changed this section to reflect some of the changes in the manuscript made in response to Reviewer 2. The section (page 24, lines 18-25) now has more emphasis on some of the general issues around tritium dating.

Figure 1. Units are missing in the legend for elevation.

These have been added and the definition of AHD (Australian Height Datum) has be added to the caption for Fig. 1.

Figures 4 and 6a: Correct the axes labelling

For some reason hidden layers (from Adobe CS) became visible in these figures following uploading (they were not visible in the files that were uploaded or .pdf's that I generated from these on my computer). We have removed the hidden layers in the final version of the figures.

Reviewer 2

The paper presents a comparison between mean transit time, MTT, between peatlands and eucalyptus forest in the Australian Alps. The authors used tritium as a tracer to model MTT using a lumped parameter approach. The authors also integrate geochemistry data and water stable isotopes in the analysis to yield important interpretations related to water storage and availability in these ecosystems. The paper is relevant to the scientific community because it provides information about underrepresented ecosystems with scare hydrologic data. The paper could eventually make an important contribution to the scientific literature. However, I believe there are issues with the paper structure and writing style. Thus I recommend major revisions before it can be considered for publication.

One of the reviewer's major comments relates to the structure of the paper. HESS does not have a fixed house style and looking through recent issues, the structure that we have used is one that is common but not ubiquitous. As outlined below in response to the specific comments, we made some of the suggested changes to the style of the paper.

I provide below some specific comments and suggestions:

Abstract: Lines 1-2: the first sentence sounds redundant (use of word "that"). Please rewrite.

The second "that" has been changes to a "which" (Page 2, line 2).

Line 8: define the acronym before using it (TU).

We defined "TU" on first usage both in the abstract (page 2, line 7) and the main text (page 4, line 11).

1. Introduction:

I believe the introduction should provide more information about the use of geochemical data the context of mean transit time modelling.

Page 3, line 21-23: This sentence should be written.

We clarified this sentence (Page 3, lines 21-23). It now reads "If water transit times are short, then the peatlands may dry significantly during droughts making them prone to degradation and vulnerable to bushfires."

Page 4, section 1.1: I think the authors should include a paragraph about the recent advances and challenges in the determination of transit times and the use of lumped parameter models. For instance, the authors should clarify that the calculated times are most likely representative of base flow conditions and spell out the underlying assumptions in the use of these models. There is a vast new literature dealing with time variant modelling of transit times.

We have addressed these comments as follows:
- We now include a discussion of both non-steady state conditions (page 5) and aggregation (page 6) that references the studies of Kirchner (2016, Hydrology and Earth System Sciences, 20, 279-297), Kirchner, (2016, Hydrology and Earth System Sciences, 20, 299-328), and Stewart et al. (2016, Hydrology and Earth System Sciences Discussion, doi:10.5194/hess-2016-532)
- We have also added a section point out the differences in approach in determining mean transit times using a time-series approach vs. individual $^3$H analyses (pages 5-6). The use of single $^3$H measurements does not require an assumption of time invariance but importantly cannot be used to determine the most appropriate lumped parameter model via comparison between the observed and predicted $^3$H activities (i.e., one has to assume a suitable lumped parameter model). This essentially utilises $^3$H in a similar way to $^{14}$C and other radioisotope tracers (which we note on page 5, lines 18-19). From a practical point of view it would be difficult to commence time-series $^3$H studies in the southern hemisphere due to the much lower activities of the bomb pulse (as we have noted on page 5, lines 24-28).

As noted below a, section discussing the impacts of aggregation on the calculations appears on pages 19-20 and Figure 8 (new).

We have noted that we are characterising baseflow conditions (page 7, line 13) and reiterated this point on page 16, lines 13-16.

Page 4, line 19-22: there is a missing "the" before "use". In addition are there any relevant references to this statement?

We corrected the typo, and following the suggestion of Reviewer 1 have referenced the Timbe et al. (2015) study (page 4, line 23).

Page 4 line 19: consider using "to determine" instead of "to determining"

This was changed (page 4, line 25).

Page 6: I suggest you eliminate section 1.3 and have the objectives of the study be the last paragraph of the introduction.

We elected to keep the section heading as it separates this material from the more general discussion of mean transit times.

Page 6 line 11-17: this information should appear before the paragraph with the objectives. Could be part of the last paragraph in page 5.

We removed this paragraph and incorporated the material in the description of the field area (page 9, lines 13-19)

I suggest this section be part of Methods.

We have made this section the first part of the Methods (section 2.1) as suggested.

Page 6 line 19-22. I suggest you rewrite this sentence.

We changed this to "Water from streams draining peatlands and eucalypt forests was sampled …" (page 7, lines 22-23).

Page 7 line 1-2, line 13-14:  Please use the correct notation for the scientific name of species.

We italicised the species names throughout section 2.1.

3. Methods

Line 9: Explain "aggregated" over what time frame? In addition, how many, when, how frequent were the samples collected?

We have clarified this to "Samples of rainfall representing the total precipitation over periods of several weeks to months (Table 1) were collected from a rainfall collector at Mount Buffalo (Fig. 1)" (page 9, lines 21-22) and also added a note regarding sample collection to Table 1.

Line 9-10: That just means grab samples, right?

We have used the term "Grab samples" (page 9, line 22).

Page 9 section 3.2.  More information is required about the modeling procedure, how were the best parameters identified, what objective function was used, how many possible parameter combinations were implemented, can you include dotty plots?  How did you chose among the 3 different transit time functions (exponential, piston flow, and dispersion). Why did you chose these 3 and not others?

As discussed above, to calculate mean transit times from single $^3$H measurements one has to assume a LPM model (i.e. it cannot be constrained from the observed $^3$H measurements in the same way as when time series data are used). This has been made more explicit in section 1.1 (page 5) and we have reiterated it here (page 11, lines 14-21). Our choice of models spans those most commonly used in the literature and are ones that accord with our understanding of the flow systems. For example the exponential-piston flow model is appropriate for near vertical recharge through the unsaturated zone and exponential flow in the unconfined shallow aquifers and soils. Additionally, where $^3$H time series data are available, these models have reproduced the $^3$H activities of river water and groundwater (e.g. Maloszewski and Zuber, 1982; Maloszewski et al. 1983; Zuber et al., 2005; Stewart et al. 2007; Morgenstern et al., 2015).

Unfortunately, there is no procedure to identify the best parameters from the $^3$H (or other data). Our approach (which is hopefully now explicit throughout) is to use a range of lumped parameter models and to recognise that this introduces uncertainty into the estimates of mean transit times (which we discuss in Section 4.2).

4. Results

Here is where the major structural issues arise:  The results section does not present any of the MTT related findings. This is odd considering that this is precisely the main topic of the paper.  Residence time results are mentioned in the abstract, discussion, and conclusions.   The manuscript must include a residence time results section in which the modelling findings are presented.

The paper is structured such that it presents the data first (Section 3) and then discusses it (Section 4). While this is not a strict requirement in HESS, it is a common structure used by many journals. While it is possible to write papers that mix results and interpretation into a single section, the drawback of doing this is that the reader may be unclear as to what is data and what is interpretation or that the paper begins to present interpretations ahead of describing the data that was used to make those interpretations. Much of what precedes section 4.2 is required to produce the interpretations in that section.

For those reasons we have kept the structure as is but have provided a brief explanation as to what is in each section at the start of Sections 3 and 4.

Page 11 lines 2-5: This sentence is awkward. Please rewrite.

We have rephrased this to "Watchbed Creek drains the peatlands at Falls Rocky A (Fig. S1b), and there is a near-complete streamflow record for this site between 1940 and 1986." (page 12, lines 21-23).

Page 11, line 8:  Avoid starting the sentence with a delta symbol, instead say" Water stable isotopes (18O and 2H).

A similar point was raised by Reviewer 1. We have gone through the paper to ensure that sentences do not start with symbols, numbers or acronyms.

Page 11: section 4.2. Figure 3 should be cited sooner.

We have cited this at the top of Section 3.2 (page 13, line 11).

In addition it is not clear when or how many samples were collected.

The samples are listed in Table 1 together with their sampling dates and we have now included numbers of samples in Figs 3 and 4. We have also clarified that these are individual samples in the caption to Fig. 3 and Table 1.

Page 11 line 11-12: is this slope different from the GMWL. The MMWL in Fig. 3 is very step is that correct. In addition, the text says that the samples line to the left of this line which is not true.

The data are to the left of the MMWL. The line through the samples is a best fit to the data with a slope of 5 (it is not the MMWL). We have added an explanation of this in the caption to Fig. 3 and have modified the Figure so that the MMWL and the trend line are more clearly distinguished.

Page 11 line 13: please provide the range if deuterium execs.

These were added to the text (page 13, lines 17-19).

Page 11 line 22-25: This sentence is too long.

This now reads "As is commonly the case in southeast Australia (Cartwright et al., 2012; Leaney and Herczeg, 1999), the $\delta^{18}O$ and $\delta^2H$ values of all waters including rainfall lie to the left of the global and Melbourne meteoric water lines" (page 13, lines 14-16).

Page 12 line 5: the word "and" makes no sense.

This was a long sentence that have made clearer by splitting (page 13, line 16).

Page 12 Lines 3-10: Did you test the data for normality to make sure a parametric test was appropriate?

Although there is no indication that the data are not normally distributed, the datasets are too small for the statistics to be rigorous. Rather than presenting the p values, we have now just pointed out that the $^3H$ activities between the sites overlap (page 14, lines 6-9).

Page 12 line 11-14: this sentence is too long.

We have split this into two sentences (page 14, line 9).

Discussion:

Page 13 line 19-21: Please avoid single sentence paragraphs.

The first paragraph to section 4 is a single sentence, but it introduces the section and we have retained it.

Page 14 line 22: reword sentence.

We split this sentence up, it now reads "Some peatland waters have lower $\delta^2H$ values than those of the rainfall. These potentially record recharge by rainfall that has a high proportion of low $\delta^2H$ stratospheric moisture. Such rainfall would also be expected to have higher $^3H$ activities than average rainfall" (page 16, lines 18-21).

Page 15: The MTT results should be move to the result section. Also the selection of the best model should be justified both physically and statistically. What objective function was use to qualify the goodness of fit?

As discussed above, we have elected to keep this material here as it is an interpretation. In terms of the approach to calculating mean transit times, as discussed above we have stressed that it is not possible to select a best model in a similar way to the way that it can be done using time-series data (this is has now been made more explicit in sections 1.1, 2.3, and 5). As to the choice of models, we provided more information as to why we used the models in Section 2.3 and the start of section 4.2. Not being able to constrain the most appropriate lumped parameter model represents an uncertainty in the calculations and we have explicitly noted that in section 4.2 (page 18, lines 4-7 and page 19, lines 7-16) and also in Section 5 (page 24, lines 17-27). We have also added the calculated mean transit times from the different models to Table 2.

Tables: in general the captions need to be more comprehensive. For instance, the caption for Table 1 should indicate if the isotopic values averages? If so what is the time period over which they were calculated, how many samples are included, and are there metrics of uncertainty.

These data are from single samples. The error bars on the figures reflect analytical errors not sample variability (as is stated in the figure captions). We have made this clearer in the caption to Table 1.

We added more explanation to Table 2 also.

Figures

Figure 1: Please use different markets to indicate the location of peatlands and eucalyptus forest.

It is too difficult to show the outcrops of peat at this scale. As noted in the response to Reviewer 1, we have included annotated high-resolution Google Earth images that show the distribution of peatlands and eucalypt forest more clearly as a Supplement.

Figure 2b. Please use probability scale in the x-axis.

We have modified the Figure

Fig. 3 not clear what the dash (–) line is.

The dashed line is a linear best fit to the rainfall samples. We have explained this in the figure caption.

The error bars for the grab samples should be representative of the accuracy (from the analysis of duplicate samples).

This is the case (as is noted in the figure caption).

If the precipitation signature corresponds to weighted means, then the error should be weighed errors.

They were analysed as individual samples (this has been added to the caption of Table 1).

Please add the number of samples (n=xx) associated to each (peat, Eucalypts, rain)

We have added this to the figure.

Figure 4:  Not sure what the equation and R2= 0.89 mean versus the R2=0.69.  The legend has a "series 3" and "linear (series 3)", automatically generated from excel, that are not identified.

As explained above, some hidden layers appeared in the final versions of these figures when they were uploaded. We have removed them.

[revised manuscript text omitted]

---

## Author Response (AR2)

(1) I strongly disagree with your statement that what you refer to as "time series approach" to estimate transit time distributions and mean transit times requires the assumption of time invariance. It is true that most lumped, convolution integral models work that way. Yet, this is not true for all; see for example Weiler et al., 2003, who introduce weighing factors to account for variable flows, or of course the original work by Niemi (1977) and his concept of flow time. In addition, there was considerable progress over the past years, explicitly allowing for variable flow conditions also in lumped models (e.g. Botter et al., 2010,2011; Benettin et al., 2013; Rinaldo et al., 2015).

We have clarified this point to:

"However, constraining lumped parameter models using sequential $^3$H data collected over several years has commonly made the assumption that the data can be described by a single lumped parameter model and that the flow system is time invariant, which may not be the case in reality (e.g., Kirchner,2016b)" (Page 5, lines 11-14).

(2) I agree that testing several models is a good idea when insufficient information about the actual system is available. This, however, does not answer the question raised by Reviewer #2: How did you choose for the different functional shapes of transit time distributions (e.g. epm, em, dm) the associated transit times (tau or taum) which you need in your eqs.1-3. This is not clear from the text and needs to be explained to allow the reader to completely follow what you are doing, because surely different values of tau can produce the observed signal with different recharge activity and different values for the other parameters. In other words, as far as I understand, you vary the recharge H3 activity and the other parameters but not tau. This needs to be clarified and justified.

We have clarified this point, we now specify:

"In this study mean transit times were calculated by matching the predicted $^3$H from the lumped parameter model to the measured value at the time of collection using TracerLPM. For these calculations, the form of the lumped parameter model and its parameters (i.e. the EPM ratio or $D_p$) and the $^3$H input function were specified" (Page 12, lines 17-20).